# WHY ATTENTION FAILS: THE DEGENERATION OF TRANSFORMERS INTO MLPS IN TIME SERIES FORECASTING

## ABSTRACT

Transformer-based architectures have achieved high performance in natural language processing and computer vision, yet many studies have shown that they have not demonstrated a clear advantage in time series forecasting. However, most of these studies have not thoroughly investigated the reasons behind the failure of transformers. To gain a deeper understanding of time-series transformers (TST), we designed a series of experiments and observed a degeneration phenomenon in the attention mechanism, which means that the attention mechanism does not contribute to the model's performance, causing the model before the final flatten head to become a stack of residual MLPs. We further conducted experiments on an interpretable dataset and found that the failure of representation learning is the cause of this degeneration of the attention mechanism. Finally, we provided a comprehensive analysis to explain why the current embedding methods fail to allow transformers to function in a well-structured latent space, and further analyzed the deeper underlying causes of the failure of embedding.

## 1 INTRODUCTION

Time series data are ubiquitous in today's data-driven world. Time series forecasting based on historical data has been a long-standing task with widespread applications across various fields, including traffic flow prediction, energy management, and financial investment. During the past few decades, time series forecasting methods have undergone significant development, from traditional statistical models (Ariyo et al., 2014) to machine learning methods (Friedman, 2001), and more recently to deep learning-based approaches (Lai et al., 2018; Liu et al., 2021; Bai et al., 2018).

Transformer (Vaswani et al., 2017) has undoubtedly become one of the most successful architectures for sequence modeling, achieving exceptional performance in various fields such as natural language processing (NLP), speech recognition, and computer vision. Giving the success of Transformer in different domains, researchers have recently begun to apply it to multivariate time series forecasting, treating each timestamp as a token embedded in the model, such as Informer (Zhou et al., 2021), Autoformer (Wu et al., 2021), Pyraformer (Liu et al., 2022) and FEDformer (Zhou et al., 2022). Due to the limitations of timestamp token models, researchers have subsequently proposed Crossformer (Zhang & Yan, 2023), PatchTST (Nie et al., 2023), which divides the sequence into patches, and iTransformer (Liu et al., 2024a), which directly treats entire channels as tokens.

However, despite the progress made by these Transformer-based methods, their effectiveness in time series forecasting, particularly for long-term time series predictions, remains a subject of ongoing debate. Zeng et al. (2023) found that simple linear models could outperform these transformer-based approaches, opening new avenues for research into simpler architectural frameworks. Recent studies by Zhang & Yan (2023) and Nie et al. (2023) reveal that the approach of directly treating timestamps as tokens hinders the attention mechanism in effectively capturing temporal patterns.

However, these studies do not fully explain the underlying reasons for the failure of Transformers, as they neither conduct an in-depth investigation into the intrinsic behavior of the attention mechanism nor provide any fundamental solutions. Therefore, the goal of this work is to further examine the limitations of attention mechanisms in time series forecasting and to theoretically explore the root causes of these problems.

Table 1: A straightforward instance of ablation experiment, in which the performance of both models remains unaffected, calls into question the efficacy of the attention mechanism.

| Model | Dataset | Multi-Head Attetion | MSE |
|---|---|---|---|
| PatchTST | ETTm2 | w/ | 0.277 |
| PatchTST | ETTm2 | w/o | 0.277 |

Building upon prior research, we further investigate patch-wise and channel-wise Transformers, extensively surveying various Transformer models, including time-series foundation models. We conduct an in-depth study of the common issues in their attention mechanisms, designing multiple experiments from different perspectives to reveal the phenomenon of Transformers degenerating into multilayer perceptrons(MLP).

We also compare PatchTST with the Vision Transformer (ViT) (Dosovitskiy, 2020)and design a toy dataset to explore the challenges faced by the attention mechanism. Through both theoretical and experimental approaches, we argue that the current embedding methods are insufficient to provide an appropriate latent space for the attention mechanism.

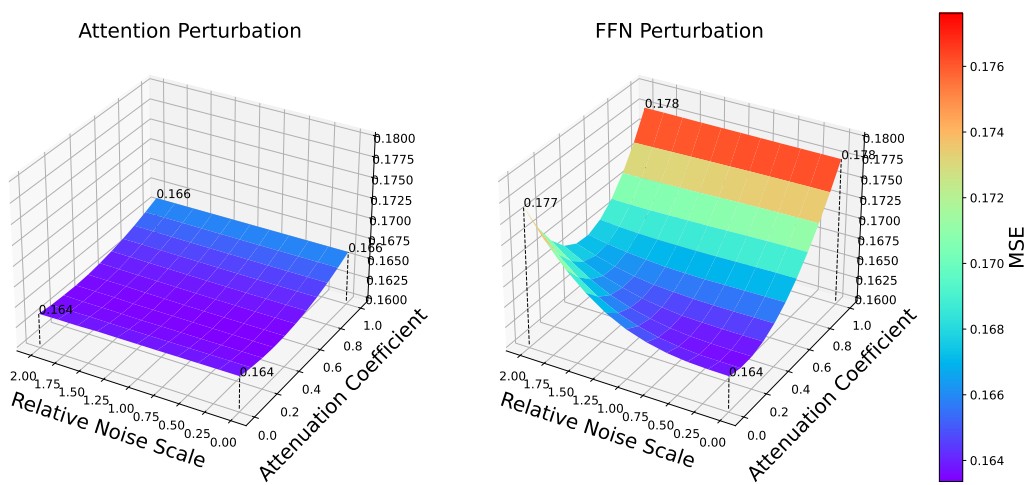

Figure 1: Results of module perturbation experiment. The x and y axes represent the strength of the two types of perturbations, while the z-axis corresponds to the model's MSE on the test set. For the FFN module, as the perturbation strength increases, the model's performance deteriorates significantly. In contrast, for the attention module, the impact of different perturbation strengths on model performance is minimal. Even when the outputs of the attention module are completely altered, the model is still able to make accurate predictions on the sequence. This raises questions about the effectiveness of the attention module.

The key contributions of our work are summarized as follows:

- We discovered the phenomenon of transformers degenerating into MLPs, and designed a series of experiments from various angles to validate this phenomenon on multiple time-series transformer models including time series foundation models.

- We designed a toy time series dataset to study the attention mechanism in a more interpretable way, proposing that the attention mechanism is not working in the expected way. Through the comparison between the original model and the Oracle embedding, we verify that current embedding methods do not provide a well-structured latent representation.

- We proposed that current embeddings are neither effective nor necessary and validated this hypothesis through ablation experiments. We found that the rough embedding leaves the transformer blocks to perform their own representation learning, and further analyzed the deeper underlying causes of the failure of embedding.

## 2 Related Work

**Time-series Transformer Models** *Time-stamps as Tokens*: Early Transformer models treated each timestamp as a token. Informer (Zhou et al., 2021) aimed to model the temporal dependence between individual time steps in the time series sequence. Autoformer (Wu et al., 2021) drew inspiration from traditional time series analysis, such as decomposition and autocorrelation, to enhance the model's ability to capture periodicity and trends. FEDformer (Zhou et al., 2022) incorporated a Fourier-enhanced structure, making it linearized. *Patches as Tokens*: To address the limitations of timestamp-level tokenization, patch-based time-series transformer architectures have gradually gained attention. Crossformer introduced a cross-dimension interaction mechanism, capturing dependencies through a patching and aggregation strategy. PatchTST divided time series into patches and treats patches as tokens to capture higher-level temporal features. *Channels as Tokens*: iTransformer (Liu et al., 2024a) treated each channel as an independent token, which has achieved significant success in enhancing multivariate time series modeling. *Time series foundation models*: Recently, there have been many time series foundation models trained on new large datasets, which predict sequences with high accuracy in a zero-shot setting. Relevant models include Moirai (Woo et al., 2024), TimesFM (Das et al., 2024), and lag-llama (Rasul et al., 2023).

**Different Voices about Time-series Transformer Models** Although the temporal Transformer has become very popular, Zeng et al. (2023) discovered that linear networks can be comparable to or even outperform Transformers in multivariate long-term forecasting. SAMformer (Ilbert et al., 2024) suggests that attention is the main reason for Transformer models' poor generalization ability, causing them to converge to sharp local minima. PITS (Lee et al., 2024a) achieves performance beyond traditional Transformers through the simple patch-wise MLP that embeds each patch independently. Kim et al. (2024) reevaluated the effectiveness of self-attention for time series forecasting by eliminating self-attention and utilizing a cross-attention mechanism. Existing research predominantly focused on timestamp token models or merely highlighted the performance limitations of Transformers. In contrast, this paper delves deeper into the underlying causes of this phenomenon, providing a more comprehensive analysis.

**Computer Vision Model** Vision Transformer (ViT) introduced a Transformer architecture to process images by splitting the image into patches, and has achieved breakthrough results in image classification tasks. It inspired the research of PatchTST which shares almost the same architecture as ViT, whereas PatchTST encounters the problem of degeneration. To investigate the difference, we performed a comparative analysis of them in the following sections.

## 3 The Degeneration of Transformers into MLPs

Since many studies have shown that the performance of transformers may not outperform linear models, we aim to design experiments to investigate the specific role of the attention mechanism in the model. We approximated the attention module through various methods, reducing it to simple summation or a token-wise linear layer, and observed no significant change in the model's performance. Furthermore, by progressively increasing the patch length, we approximated the model to a single token input, effectively reducing the model to just the FFN layers, and once again, the model performance remained unchanged! This indicates that the current attention mechanism fails to effectively capture and analyze contextual relationships and does not play a significant role; its contribution to the model's performance is minimal. Its sole function appears to be mixing different tokens together, but it does not learn meaningful mixing weights. **The contribution of multi-head attention to the overall model performance is negligible, and the transformer block is essentially only the FFN module at work, causing the model backbone to degrade into an MLP.**

### 3.1 Attention Replacement Experiment

Our first experiment is an ablation study of the attention mechanism. The central idea is to replace the attention matrix in the transformer blocks with certain matrices and assess the impact of attention by observing the changes in model performance. Four distinct strategies are employed: setting the attention matrix to a zero matrix, an identity matrix, an average matrix, or a fixed but trainable attention matrix. Zero attention is actually an ablation experiment for the attention mechanism. And other matrices are the transition between the zero matrix and the original matrix.

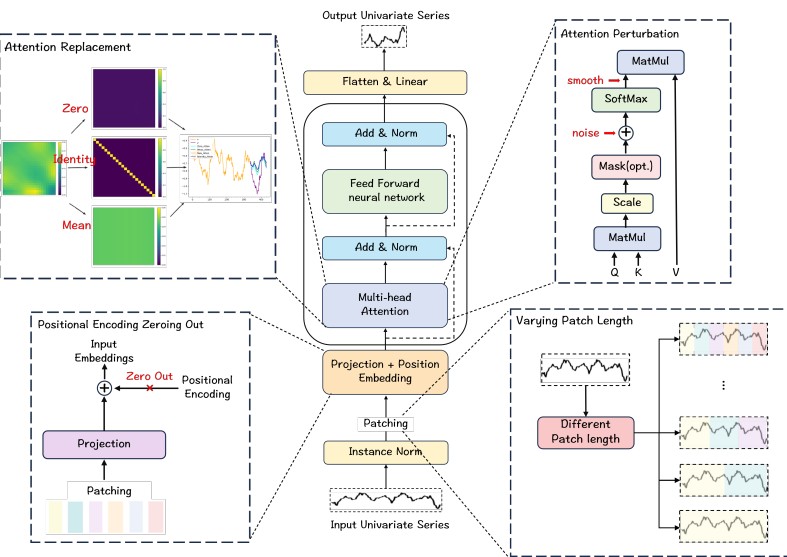

Figure 2: Experimental setup. This figure illustrates the four experiments conducted in this section.

In the Transformer architecture, Multi-Head Attention first projects the input tokens through three linear layers to obtain the queries ($\mathbf{Q}$), keys ($\mathbf{K}$), and values ($\mathbf{V}$). These are then used to compute the attention via the Scaled Dot-Product Attention mechanism.

$$\text{Attention}(\mathbf{Q}, \mathbf{K}, \mathbf{V}) = \mathbf{A}\mathbf{V} = \text{softmax}\left(\frac{\mathbf{Q}\mathbf{K}^\top}{\sqrt{d_k}}\right)\mathbf{V} \tag{1}$$

When the attention matrix $\mathbf{A}$ is set to zero, the matrix on the right-hand side of Equation (1) essentially becomes a zero matrix. Since Multi-Head Attention is followed by a residual connection, the Transformer block essentially consists only of the Feed-Forward Network (FFN) module. As a result, the model backbone is effectively reduced to an MLP. The FFN operates on individual tokens, causing each block to lose its ability to aggregate information across tokens. For PatchTST, different tokens remain in an independent state until the final flatten and linear layers aggregate them. In the case of iTransformer, zero attention is equivalent to a channel-independent MLP, where there is no information exchange between channels.

Since the ablation study is conducted on the same model architecture, setting attention to zero leads to the failure of the three linear layers in the multi-head attention module, resulting in a significant reduction in the model size. Using an identity attention matrix mitigates this issue while ensuring that the tokens remain mutually invisible. Additionally, average attention refers to setting each element of the attention matrix to a constant value of 1/T, where T is the number of tokens. Average attention does not utilize any contextual information; it simply combines all the value ($\mathbf{V}$) vectors, makes tokens visible to each other, and this visibility is not derived from analyzing context. In addition to this, we also conducted experiments with fixed but trainable attention, which behaves like an extra linear layer after the $\mathbf{V}$ matrix at the token-wise level, whose weights are priors and context-independent, similar to the approach used in TSMixer (Chen et al., 2023).

We conducted our experiments on PatchTST and iTransformer, with the results shown in the figure above. Detailed result tables and parameter information are provided in the appendix. **Remarkably, the model's performance remained unaffected even after the replacement of the context-aware attention mechanism.**. For all attention variants, the model's performance remained largely unchanged across most datasets, with performance even improving on certain datasets. On the ECL and Traffic datasets, the zero-attention model experienced a slight performance drop, but average and trainable fixed attention maintained stable performance. These results undoubtedly raise questions regarding the effectiveness and necessity of the attention module. We conducted the same experiment across multiple models including time series foundation models, and the results were consistent with those of the aforementioned experiments; further details can be found in the appendix.

## 3.2 ATTENTION PERTURBATION EXPERIMENT

To further investigate the impact of the attention module on the performance of the network, we conducted a perturbation experiment on a **trained model**. We applied identical perturbations to both the multi-head attention module and the feed-forward neural network (FFN) module and analyzed the resulting changes in network performance. The perturbation formula is as follows:

$$\mathbf{A} = (1 - \alpha) \cdot \mathrm{softmax}\left(\frac{\mathbf{Q}\mathbf{K}^\top}{\sqrt{d_k}} + \epsilon_{T \times T}\right) + \frac{\alpha}{T} \cdot \mathbf{1}_{T \times T} \tag{2}$$

$$\epsilon_i \sim \mathcal{N}(0, \sigma_i^2 \mathbf{I}) \tag{3}$$

$$\sigma^2 = \eta \cdot \mathrm{Var}\left(\left(\frac{\mathbf{Q}\mathbf{K}^\top}{\sqrt{d_k}}\right)_i\right) \tag{4}$$

In these equations, $\alpha$ represents the attenuation coefficient, and $\eta$ denotes the relative noise scale. $\mathbf{A}$ is the resulting attention matrix. To ensure numerical stability, we utilize $\alpha$ to balance between the mean attention and the original attention, and noise is introduced prior to the application of the softmax function. The perturbations due to attenuation and noise disrupt the attention mechanism in distinct ways: when $\alpha = 1$, the attention mechanism reduces to a simple mean summation; and when the noise level is excessively high, the attention mechanism degenerates into a random combination of tokens. For the FFN perturbation, noise is introduced before the activation layer, controlled by $\eta$, and the output is subsequently smoothed by the parameter $\alpha$.

The experimental results are shown in Figure 1. Even after the model had been trained, the experiment still demonstrated that the degeneration of the attention mechanism did not cause any noticeable disruption to the model. In contrast, perturbing the FFN module led to a substantial decrease in model performance. This suggests that the model's performance is concentrated in the FFN, and that the attention mechanism fails to play its intended role.

## 3.3 VARYING PATCH LENGTH EXPERIMENT

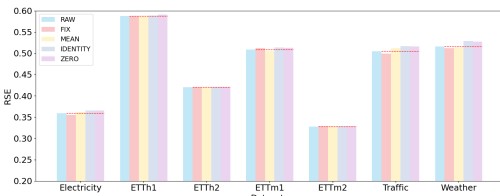 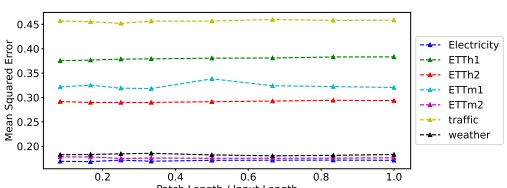

Figure 3: Left (a) Results of attention replacement experiments. The legend represents different types of attention. Right (b) Varying patch length experiment. We conducted the experiment on PatchTST with the stride equal to the patch length, minimizing the model size deduction due to growing patch length.

In this subsection, we also examined the effect of varying patch lengths on model performance. As the patch length increases, the number of tokens gradually decreases, and the scope of the attention mechanism narrows. When the patch length equals the length of the output sequence, the input reduces to a single token, causing PatchTST backbone to degenerate into a simple MLP. The experimental results in Figure 3b indicate that the performance is almost unaffected by changes in patch length. Even when the patch length is equal to the input length, and the model backbone is effectively equivalent to an MLP, there is no significant decrease in performance, which calls into question the role of the attention mechanism.

## 3.4 POSITIONAL ENCODING ZEROING OUT EXPERIMENT

Since the attention mechanism is permutation-invariant, it is crucial for patch-wise models to use positional encoding to preserve the temporal positional information between tokens. If this positional information is lost, the model will be unable to handle temporal dependencies, except for

the final flatten and linear layers. We studied and compared PatchTST and ViT, and found that the importance of positional encoding varies significantly between the two models. By zeroing out the positional encodings in both **trained** models, we observed a sharp performance degradation in ViT, while PatchTST's performance remained unchanged. PatchTST adds a flatten layer and a linear layer after the transformer blocks, where the relative positions of the linear layer weights corresponding to different tokens are fixed. As a result, the model is able to recognize the relative positions of the tokens even in the absence of positional encoding.

Table 2: Positional encoding zeroing out experiment.

| Model | Metric | PosEnc | Zero PosEnc |
|---------|----------|--------|-------------|
| ViT | Accuracy | 89.8% | 32.4% |
| PatchTST | MSE | 0.142 | 0.142 |

We quantified the similarity between encodings at different positions and found that, for ViT, the similarity between adjacent positional encodings was higher. In contrast, for PatchTST, there was no such correlation between position and similarity. This suggests that, for PatchTST, the attention mechanism does not utilize any positional information, which is especially critical for time series data.

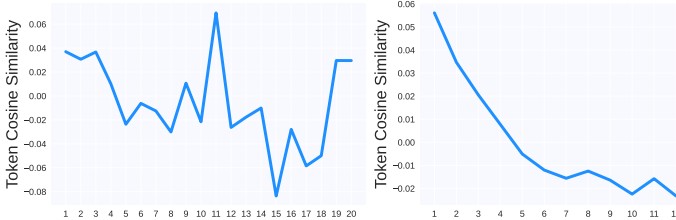

Figure 4: Relationship between token similarity and token distance. Overall, ViT exhibits a downward trend. An outlier is observed at a distance of 14, likely caused by the limited number of samples at this distance, as only two pairs of patches have a distance of 14.

## 4 FAILURE TO CAPTURE INTER-PATCH DEPENDENCY

To further analyze the attention mechanism and explore the underlying causes of the degradation, we study the attention to dependencies between different patches. We designed a more interpretable toy dataset and performed experiments on it, finding that the attention mechanism is not working in the expected way and fails to capture inter-token dependency due to the poorly structured latent space.

The core idea of the attention mechanism is to dynamically assign different "weights" or "attentions" according to the interaction between different tokens. This mechanism enables the model to be more efficient when processing long sequences, focusing on key parts, and preventing information loss. For time series data, the attention mechanism can capture relationships between different patches, model the context, and make final predictions based on prior knowledge and posterior attention.

### 4.1 A MORE INTERPRETABLE TOY DATASET

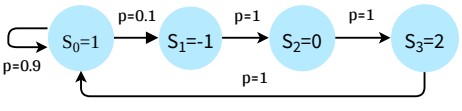

Figure 5: The state machine used in our toy dataset.

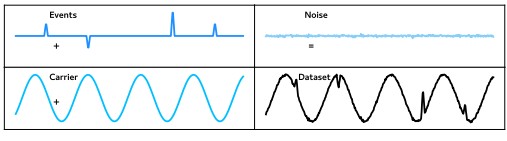

Figure 6: Toy dataset composition.

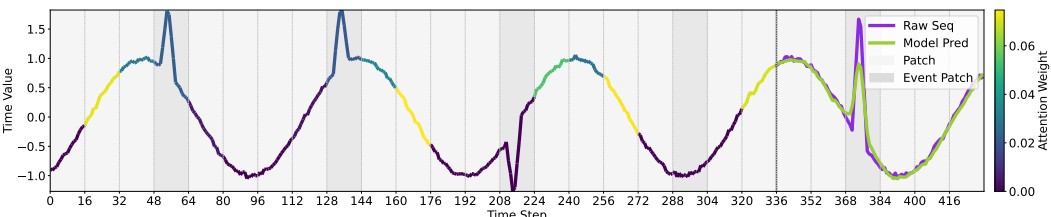

Figure 7: Toy dataset attention visualization. The figure depicts the attention distribution of patch [336-351] over the input sequence, with weights quantified by color intensity. The attention on the actual event patches is surprisingly low, with the attention on the most recent event patch with $S_2 = 0$ being almost zero! This results in the model failing to successfully predict the next event with $S_3 = 2$, erroneously estimating it as $S_0 = 1$.

Common public datasets in time series are highly specialized, with too much complexity and variability, making it difficult to quantify the relationships between patches. To address this, we designed a toy dataset based on state machines. We trained PatchTST on it and analyzed the attention results. The dataset consists of three parts: the carrier wave, the event signal, and the noise. The carrier wave is a simple sine wave. The event signal is a periodic triangular hat waveform whose amplitude is controlled by a state machine. Within each cycle, the system is in a state $S$, and the signal waveform for that cycle is controlled by $S$. The state of the next cycle is determined solely by the current cycle. Figure 5 illustrates the state machine used in the experiment.

## 4.2 TOY DATASET ATTENTION EXPERIMENT

An ideal transformer model should allocate sufficient attention to the patch where the event occurs, especially the most recent event. It should then predict the state of the next event using the most recent event state and the learned state machine model.

We performed experiments on PatchTST, with the experimental details provided in Section G.2. However, the result provided in Figure 7 showed that the model did not learn the state machine model, nor did it focus on the event patches. This indicates that in current time-series transformers, the attention mechanism may not effectively understand the information contained in the tokens, nor does it analyze their relationships based on the token data. In our toy dataset, during representation learning process, the model did not express the concept of "events," making it difficult for the attention mechanism to function properly. This also explains why, in previous attention replacement experiments, we found that context-dependent real-time attention was dispensable for the model. In fact, once we use the Oracle embedding in Section B to manually introduce semantic information, the attention mechanism is then able to correctly capture the relationships between events, thereby producing the correct predictions. This leads us to reconsider how the model embeds time series data into the latent space.

## 5 INEFFECTIVENESS OF CURRENT EMBEDDING METHODS

Based on the previous analysis, the degradation may result from suboptimal representation learning, and representation learning in Transformers is initially carried out by the embedding layer. Current time series Transformers primarily employ linear layers as embedding layers. In this section, we study the effectiveness of linear embedding and prove experimentally that the current linear embedding method fails to offer a well-structured latent space for transformer blocks to function in, making itself neither effective nor necessary for time series data. Due to space limitations, a deeper discussion in Section A will reveal that the difficulty of representation learning arises from the characteristics of TST data, which render mainstream embedding methods—including linear embedding—ineffective.

## 5.1 THEORETICAL ANALYSIS OF LINEAR EMBEDDING

Transformers operate within a latent space, where the attention mechanism captures the relationships among latent vectors corresponding to different tokens. These relationships are then leveraged to refine and optimize the latent vectors, enabling them to encapsulate richer and more global information. The role of the embedding layer is to project the input data into this latent space, which encodes the inherent properties of the input and provides prior information. For time series data, the embedding layer is responsible for mapping the time series into an appropriate latent space. This space characterizes the relevant attributes of the waveform and may also include representations of the physical phenomena underlying the temporal data.

From the simplest perspective, in a simplified scenario, as a commonly used implementation of embedding layers in time series Transformers, a full-rank linear layer with the same input and output dimensions is an isomorphic linear transformation. **However, the latent space, which may encode high-level data such as semantic, structural, or dependency-related information, cannot be isomorphic to the time series space**.

In transformers used for natural language processing (NLP), the embedding layer serves to map discrete tokens into a continuous dense latent vector space, enabling the originally discrete inputs to be processed and learned within a continuous domain.

In NLP, the embedding layer is typically implemented as a linear layer, with the input represented in a one-hot encoded form that performs as a table lookup operation. Each token corresponds to a sub-vector in the weight matrix. Many Transformer models in the time series domain have adopted this approach. Table 7 presents the embedding implementations of important time series models. Most non-large models and some large models use either linear or convolutional layers for embedding, with convolutional neural networks serving as a form of weight-sharing linear layer.

In time-series transformers, the embedding layer is used to map the input time series into a continuous dense latent vector space. This latent space captures various characteristics of the time series waveform and may also include information relevant to the physical phenomena underlying the sequence. However, as a linear transformation, a linear layer may struggle to project the time series onto a high-dimensional manifold, thereby failing to produce a meaningful latent representation.

The inputs of NLP and time series forecasting are fundamentally different. In NLP, the input consists of one-hot vectors, while the input of time series is a continuous sequence of data over time. Directly adopting the linear embedding approach from NLP transformers without thorough investigation and consideration raises concerns about its validity.

Therefore, linear embedding may not be an effective method for embedding time series data into a well-structured latent space. However, time series input vectors do not reside in a linear space, and the latent space is not necessarily a linear space either. Generally speaking, we can approximate the latent vectors as lying on a high-dimensional manifold, while the time series itself can be viewed as a window sampling from a function determined by hyperparameters and noise.

## 5.2 EXPERIMENTAL EVIDENCE

Linear transformation is also an unnecessary embedding. In most models, once the input is embedded, it proceeds directly to the attention module, where the QKV matrices are computed through linear layers. This means the linear layers of attention follow immediately after the linear embedding layer. However, the simple stacking of multiple linear layers is equivalent to a single linear layer, implying that the linear embedding is unnecessary or, at best, replaceable.

Nevertheless, due to the existence of positional encoding and the dimensional transformation requirements, the embedding layer cannot be simply removed. We provide an experimental demonstration that linear embedding does not significantly contribute to model performance improvement. We conducted experiments on several models, where we froze the initialized embedding layer to prevent it from training and compared the results with the original model.

As shown in Table 25 and Table 24, when the weights of the embedding layer are fixed and excluded from training, the model's performance remains unaffected. This demonstrates that the linear layer, when used as an embedding layer, contributes negligibly to the model's performance, and its role in representation learning is not significant.

## 5.3 ViT as an exception

Although we have theoretically and experimentally confirmed that using a linear layer as the embedding layer is ineffective and unnecessary, ViT achieves impressive performance using linear embeddings. To explain this discrepancy, we designed a simple experiment.

We applied attention smoothing to a trained ViT model with 8 blocks, setting attention to the mean matrix, and observed the change in model performance. We found that smoothing in the first four blocks had a minimal effect on performance, whereas smoothing in the last four blocks led to a significant performance drop. The attention modules in the first few blocks contribute less to the update of the hidden vectors. In other words, the first four blocks primarily use the FFN for representation learning, and attention only starts to play a significant role in the latter four blocks, where the representation learning has been completed.

To check this in time series models, we gradually increased the number of blocks in PatchTST and iTransformer, hoping that the additional blocks could assist with representation learning. However, as shown in Figure 21 and Figure 22, the model performance did not improve significantly; and overfitting occurred in some models and datasets. The representation learning strategy for time series data remains a critical issue.

Table 3: Results of ViT attention smoothing. We smooth the attention instead of setting it to zero in order to maintain the numerical stability of the model. In fact, experiments with zero attention exhibit the same trend as those with smoothing, with the only difference being a lower accuracy.

| Smoothed Block ID | Accuracy | Degradation |
|---|---|---|
| None | 89.8% | - |
| 0, 1 | 88.8% | -1.0% |
| 2, 3 | 84.4% | -5.4% |
| 4, 5 | 73.2% | -16.6% |
| 6, 7 | 75.0% | -14.8% |
| 0, 1, 2, 3 | 82.1% | -7.7% |
| 4, 5, 6, 7 | 44.2% | -45.6% |
| 0, 1, 2, 3, 4, 5, 6, 7 | 33.4% | -56.4% |

## 6 Discussion

**Conclusion**  In this study, we unveil the phenomenon of time-series forecasting Transformers degenerating into MLPs and substantiate its prevalence across various models and datasets through extensive experimentation. By designing an interpretable toy dataset and leveraging visualization analyses, we reveal that the attention mechanism fails to effectively capture the critical dependencies within time-series data. Furthermore, we provide both theoretical and empirical evidence demonstrating that the current linear embedding approach is neither effective nor necessary. These findings suggest that existing time-series Transformer architectures require advancements in representation learning to construct a more expressive latent space, thereby enabling the attention mechanism to function as intended. In Section A, we further discuss the challenges in representation learning for Time Series Transformers. While our findings do not imply that Transformers are inherently unsuitable for time series forecasting, based on the comprehensive and solid analysis above, we remain cautiously skeptical about their potential in this domain.

**Future Work**  Our findings have profound implications for the future development of Transformer-based models in time-series analysis, particularly for foundational models that require substantial computational resources. We advocate that foundational models should prioritize improving representation learning before focusing on refinements to the attention mechanism itself. Moreover, the degradation phenomenon we have identified is not limited to time-series forecasting but may also extend to other domains, such as classification and imputation, necessitating further investigation. Finally, we advocate for the development of a novel metric or benchmark to assess the extent of attention's influence, as time series data, unlike those in the domains of image and language, are not as easily comprehensible to humans, rendering the interpretation of attention weights significantly more intricate.

## REPRODUCIBILITY STATEMENT

We have made extensive efforts to ensure the reproducibility of our work. An anonymous implementation of our proposed methods is available at `https://anonymous.4open.science/r/TST-Degeneration-1050`. The appendix provides detailed descriptions of the experimental setups, hyperparameters, and additional results that support the findings in the main text.

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

## A ADDITIONAL DISCUSSION

In the main body of this paper, we argue that the commonly used linear embedding layer in current time series forecasting Transformers is not an effective embedding mechanism. We focus on linear layers because they represent the most prevalent form of embedding implementation. However, further experiments shown in Table 4—which essentially covers all embedding layers currently adopted in time series Transformers—demonstrate that various embedding strategies, including linear layers, fail to avoid the issue of attention degradation. In other words, the challenges in representation learning for time series Transformers are not solely caused by the linear embedding layer, but instead stem from deeper, more fundamental issues.

Table 4: Attention replacement carried on PatchTST with different embedding layers. The dataset is weather; prediction length = 96.

| ATTENTION TYPE | LINEAR EMB | CONV EMB | MLP EMB | RESIDUAL EMB |
|---|---|---|---|---|
| RAW | 0.157 | 0.153 | 0.154 | 0.158 |
| MEAN | 0.154 | 0.159 | 0.155 | 0.155 |

In Section 5.3, we discuss the ViT model, where representation learning is jointly accomplished by the linear embedding layer and the early Transformer blocks. However, as shown in Section G.5, simply increasing the number of blocks does not reproduce this behavior in time series scenarios. In some cases, it even leads to overfitting. This discrepancy may be attributed to the substantial differences between input data in computer vision (CV) and time series (TS) domains. In CV, an image is divided into multiple patches, each of which still contains a relatively large amount of data. This makes representation learning a process of dimensionality reduction and data compression. As long as sufficient information exists within the input, it can potentially be effectively encoded. By contrast, in time series data, each patch corresponds to only a small amount of information.

For example, in the official implementation of iTransformer, the embedding layer projects input data of length 96 into a 512-dimensional latent space—effectively a 5.3× dimensional expansion. For PatchTST, the embedding layer projects 16-dimensional input data into a 128-dimensional token space, yielding an 8× increase. Even when auxiliary techniques such as contrastive learning are introduced to help shape the latent space, such a high dimensional uplift makes it inherently difficult for the model to capture intrinsic patterns in time series data.

Table 5: Input data and latent space size across different domains.

| Model | Input Size per Token | Hidden Size | Remark |
|---|---|---|---|
| BERT | 1×30522 = 30522 | 768 | bert-base-uncased, vocabulary size=30522, one-hot |
| ViT | 3×16×16 = 768 | 768 | vit-base-patch16-224, patch size=16 |
| PatchTST | 1×16 = 16 | 128 | patch size=16, channel independent |

In other models, this issue might be alleviated by reducing the dimensionality of the latent space. However, in Transformers, since the QK product involves vector multiplication, the latent space must be sufficiently large to accommodate a sufficient number of approximately orthogonal semantics. Compared to the CV domain, the actual input volume of time series data is quite small. Whether such data inherently contains—or even requires—so many semantic dimensions remains an open question.

Moreover, the "smallness" of time series data goes beyond individual input size—it also manifests in the heterogeneity of distributions across different datasets. In CV or NLP, although datasets may vary in focus, the underlying world knowledge they reflect is often consistent. For instance, datasets composed of academic papers versus novels may differ in terms of rigor, subject matter, and factual density, but they share consistent grammar and world knowledge. Even datasets in different languages tend to offer coherent representations of the world. In computer vision, despite differences between datasets like automobiles and pets, the underlying physics of texture, lighting and geometry remain consistent.

In contrast, time series datasets often exhibit completely divergent distributions. For example, financial time series and seismic activity series have fundamentally different characteristics. As a result, even for large time series models trained on diverse datasets, representation learning remains difficult due to distributional inconsistency, and attention degradation still occurs (see Table 23, Table 22 and Table 20).

Table 6: Ratio of Input to Hidden across different models.

| Model | Input Size per Token | Hidden Size | Ratio of Input to Hidden |
|---|---|---|---|
| iTransformer | 96 | 512 | 0.1875 |
| PatchTST | 16 | 128 | 0.125 |
| TimesFM | 32 | 1280 | 0.025 |
| Pathformer | 2–16 | 8 | 0.25–2 |
| SAMformer | 336 | 16 | 21 |
| ViT | 3×16×16 = 768 | 768 | 1 |
| SwinV2 | 3×4×4 = 48 | 768 | 16 |

In summary, this section highlights key difficulties in the representation learning process of time series Transformers, in the hope that future research can address and improve embedding strategies. While our findings do not imply that Transformers are inherently unsuitable for time series forecasting, we remain cautiously skeptical about their potential in this domain.

# B   VALIDATING THE HYPOTHESIS ON THE TOY DATASET VIA ORACLE EMBEDDING

As discussed above, the failure of representation learning leads to the Transformer model's failure on the toy dataset. This section verifies this hypothesis by using an Oracle Embedding.

We design an Oracle Event Embedding (OEE) that directly encodes the ground-truth event type of each patch, bypassing the representation learning stage. We add the OEE to the output of the embedding layer of PatchTST. This method does not require the embedding layer to learn the semantic information contained in the signal; instead, it directly inputs the semantic information into the model as a feature, which is equivalent to obtaining a good event representation.

In Figure 8, compared with the original embedding strategy, the model using the oracle embedding captures the relationships between events and accurately predicts the next event. In Figure 9, it can be observed that the attention mechanism now utilizes the information contained in the previous event to predict the next one, which aligns with the intuitive understanding of how attention should work.

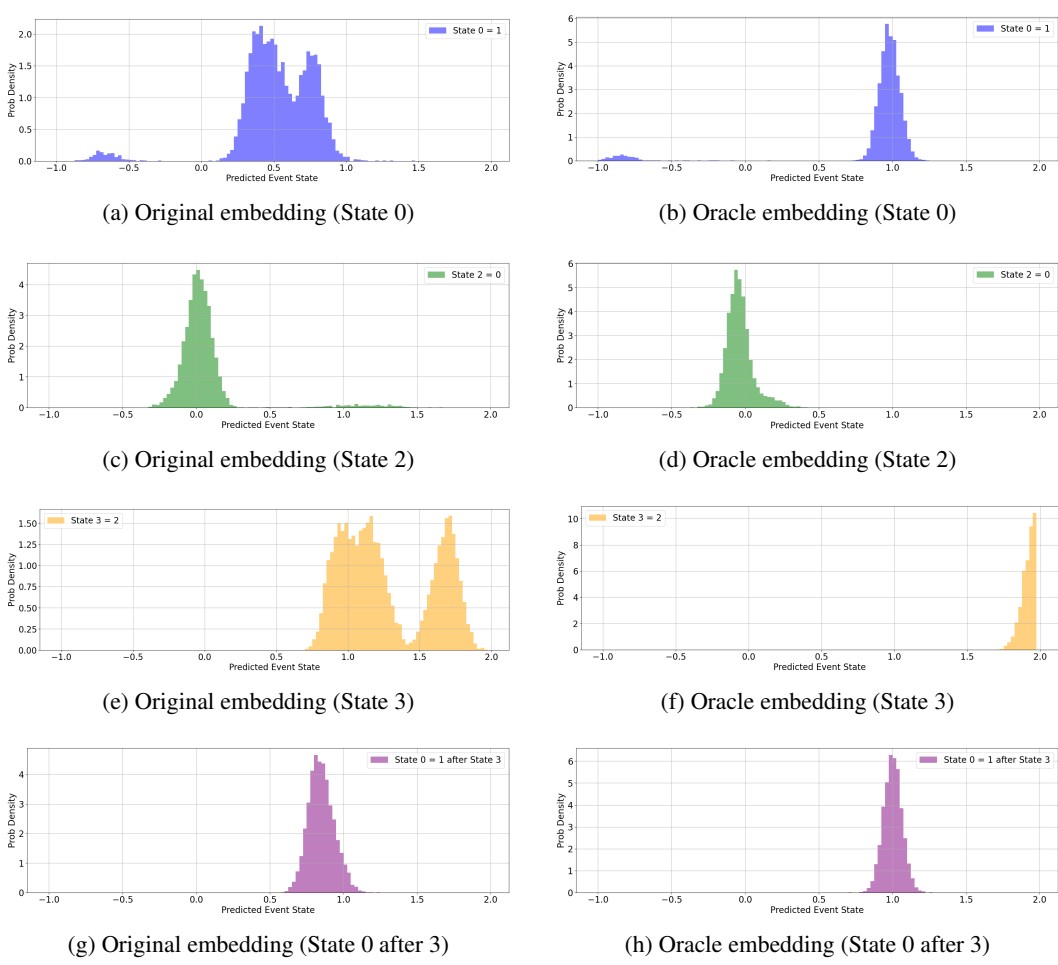

(a) Original embedding (State 0)  (b) Oracle embedding (State 0)

(c) Original embedding (State 2)  (d) Oracle embedding (State 2)

(e) Original embedding (State 3)  (f) Oracle embedding (State 3)

(g) Original embedding (State 0 after 3)  (h) Oracle embedding (State 0 after 3)

Figure 8: Comparison between oracle embedding and original embedding on toy dataset.

In Figure 8, the original embedding uses an MLP as the embedding layer, and its results are consistent with those in Figure 11, Figure 12, Figure 13, Figure 14, and Figure 15 (which use a linear layer as the embedding layer). This also verifies the discussion in Section A.

The oracle embedding used in this experiment can only be applied to the toy dataset. For real-world datasets, a new representation learning method is still required to obtain good latent representations.

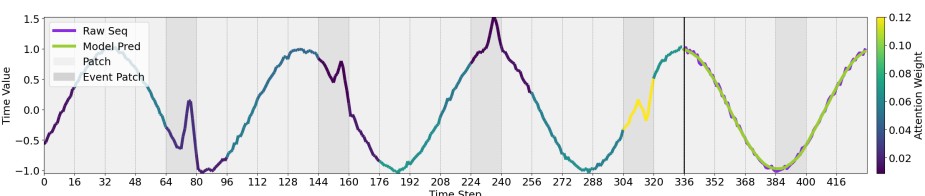

Figure 9: Attention visualization of oracle embedding.

## C  INFORMAL BRIEF DISCUSSION OF POSSIBLE SOLUTIONS

As discussed in the main paper, we believe that future improvements to time series Transformers should focus on learning better latent representations. Inspiration can be drawn from Latent Diffusion Model in image generation and speech-generation models such as Qwen2.5-Omni and LLaMA-Omni. These models typically employ an encoder-decoder architecture to extract high-quality representations of the input (image or speech), and only then apply the Transformer module to perform attention-based modeling in the latent space. In essence, these approaches place the Transformer within an already meaningful and structured representation space, allowing attention to operate more effectively.

Translating this idea to the time series domain, it suggests that building a high-quality time series encoder-decoder could be crucial to unlocking the full potential of Transformers.

One particularly promising approach is to use VQ-VAE or RQ-VAE to map the time series into a discrete codebook, which aligns well with the discrete token inputs and outputs that Transformers are designed for. For forecasting tasks, a VAE encoder can produce discrete tokenized representations as Transformer input, while the Transformer predicts a distribution over the codebook, and a VAE decoder reconstructs the output time series from the predicted tokens.

This approach also addresses a key limitation in PatchTST, where an additional concatenation and a prediction linear head (which introduce significant additional parameters) are required to aggregate all token representations and produce the final output. By adopting a discrete representation and next-token prediction paradigm, this method brings time series forecasting closer to the standard autoregressive framework used in mainstream Transformer architectures. A thorough and serious discussion and evaluation of such solutions would require substantial space and careful analysis, which is why we did not include them in the main paper.

## D  THE FORMAL NOTATION FOR THE TOY DATASET GENERATING PROCESS

We construct synthetic time series $x(t)$ as the superposition of three components:

$$x(t) = x_{\text{carrier}}(t) + x_{\text{event}}(t) + x_{\text{noise}}(t) \tag{5}$$

Carrier Wave : A sinusoidal base signal defined as below, with amplitude $A = 1$, frequency $f = 0.01$, phase $\phi = 0$, and offset = 0.

$$x_{\text{carrier}}(t) = A \cdot \sin(2\pi f t + \phi) + \text{offset} \tag{6}$$

Event Signal : A piecewise triangular waveform modulated by a discrete state machine. Each event is triggered periodically (every $T_{\text{event}} = 80$ steps) and occupies $T_{\text{event}}/r = 10$ steps, where $r = 8$. The triangle height is determined by the current state value s, scaled by an amplitude factor of 0.5. The state evolves according to predefined transition rules.

Noise : Additive white Gaussian noise

$$x_{\text{noise}}(t) \sim \mathcal{N}(0, \sigma^2), \quad \sigma = 0.025 \tag{7}$$

## E  COMPARISON WITH RESEARCH ALONG SIMILAR LINES

Several studies have put forward perspectives similar to ours, particularly Guo et al. (2024). However, there are fundamental differences between their work and ours, and we consider it necessary to provide clarification and discussion here.

Guo et al. (2024) did not identify the phenomenon of attention degradation. Their main focus was to demonstrate that the substantial computational overhead introduced by attention yields only marginal performance gains. However, their ablation studies do not imply that attention is ineffective: when the attention mechanism is entirely removed, the performance deteriorates across various models and datasets ( due to the reduced parameter count and computational complexity ), as also acknowledged in their first contribution. This only suggests that the trade-off between computational cost and performance improvement is not ideal. Importantly, since removing attention leads to performance degradation, it does not support the claim that the attention mechanism is not working, nor does it indicate that attention has degenerated.

In addition to Guo et al. (2024), Zeng et al. (2023) also put forward similar perspectives. They observed that Transformers underperform MLPs in terms of predictive performance and that positional encodings fail to contribute effectively. However, while they explained these phenomena, they did not conduct a more in-depth, fine-grained investigation into the attention mechanism itself, nor did they identify the degeneration of attention. Moreover, they did not attribute issues such as the ineffectiveness of positional encodings to shortcomings in representation learning.

In contrast, our work does not ablate the attention module directly, as this would lead to a reduction in model parameters and computational load, leading to performance degradation and confounding the interpretation of performance changes. Instead, we replace the original QK-based attention matrix with alternative forms, carefully ensuring that the computational cost and parameter count remain roughly consistent with the original model (e.g., mean and eye attention preserve the utility of the v_proj and o_proj layers, while fix attention replaces q_proj and k_proj with new parameters ). Under this setting, we observe that the model performance does not exhibit the consistent degradation reported in RAM's experiments. Instead, it remains comparable to, or in some cases even surpasses, the original model. This experimental design reveals the degradation issue in attention mechanisms.

Moreover, our work explores the degradation phenomenon from multiple perspectives, including latent perturbation, patch-length approximation, and position encoding. Compared to Guo et al. (2024)'s relatively simple ablation approach, we provide a much more systematic and multi-faceted analysis of potential issues in attention mechanisms.

A central contribution of our paper is the in-depth explanation of why such degradation occurs : why the computational cost of attention does not translate into proportional performance gains, why representation learning is important for attention, and why current time series Transformers fail to meet these expectations. These are critical aspects that Guo et al. (2024) does not address at all, but which lie at the heart of our work.

## F   SURVEY RESULTS

Table 7: Embedding strategy investigation.

| MODEL | EMBEDDING STRATEGY |
|---|---|
| AUTOFORMER | CNN |
| FEDFORMER | CNN |
| ITRANSFORMER | LINEAR |
| PATCHTST | LINEAR |
| LEDDAM (Yu et al., 2024) | LINEAR |
| CARD (Wang et al., 2024) | LINEAR |
| PATHFORMER (Chen et al., 2024) | LINEAR |
| TIMER (Liu et al., 2024b) | LINEAR |
| LAG-LLAMA | LINEAR |
| MOIRAI | MULTIINSIZELINEAR |
| TIMESFM | MLP |
| CHRONOS (Ansari et al., 2024) | RESIDUALBLOCK |

## G   EXPERIMENTAL DETAILS

### G.1   A SAMPLE OF ATTENTION REPLACEMENT

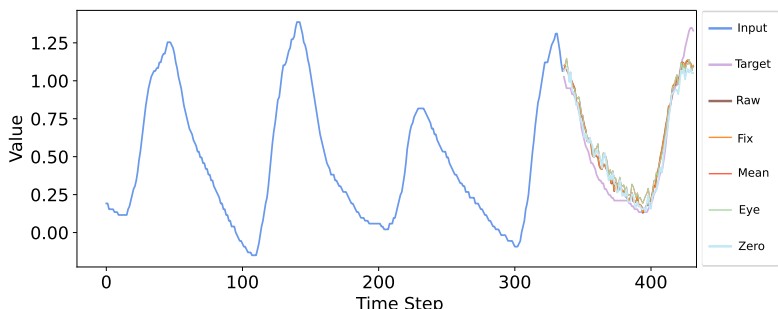

Figure 10: Comparison of time series predicted by different attention replacement methods. The legend represents different types of attention.

### G.2   TOY DATASET ATTENTION EXPERIMENT

In Figure 11, Figure 12, Figure 13, Figure 14 and Figure 15, we can observe the model's prediction distributions across different states, which are derived from 51,200 samples. The x-axis represents the amplitude of the predicted triangular hat event wave, obtained by subtracting the original carrier wave from the output waveform and integrating the residual signal.

Since the transition from State 0 to State 1 is random, the poor prediction performance for these states is expected. However, even though the occurrence of State 2 is deterministic, the model still exhibits a small probability of misclassification. The performance for State 3 is even worse. According to Figure 14, the predicted amplitude of State 3 is often biased toward 1. Additionally, in some cases, the model seems to oscillate between 1 and 2, leading to a final prediction fluctuating around 1.75. These statistical findings strongly indicate that the model fails to learn the underlying state machine and does not effectively leverage contextual information to predict the next event state.

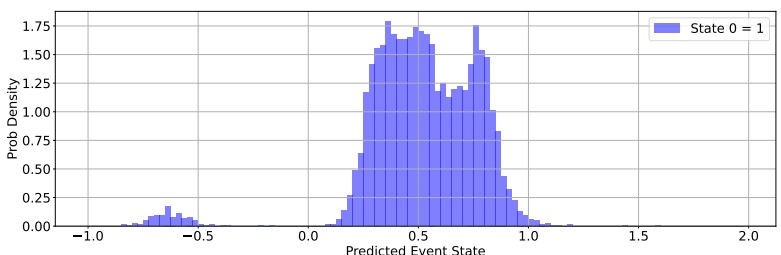

Figure 11: Prob density of predicted event state 0

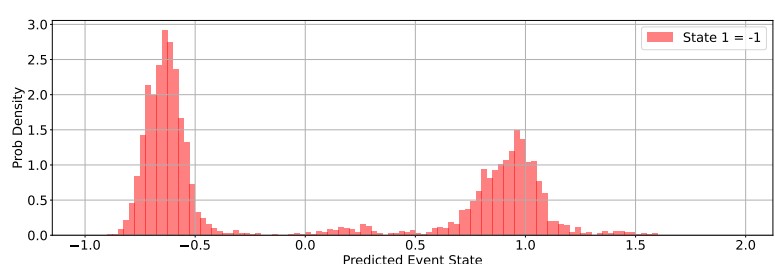

Figure 12: Prob density of predicted event state 1

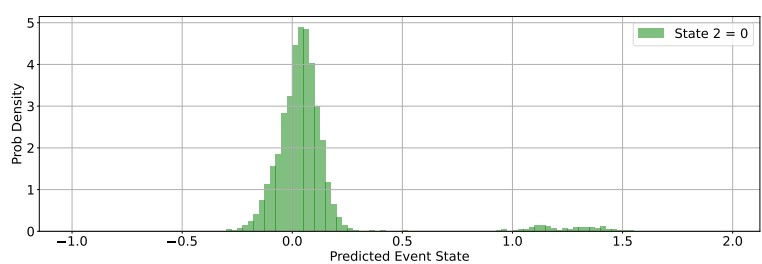

Figure 13: Prob density of predicted event state 2

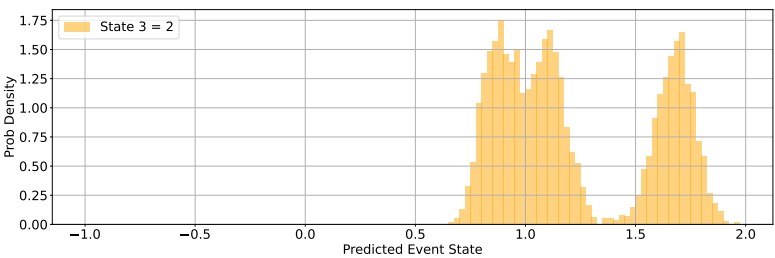

Figure 14: Prob density of predicted event state 3

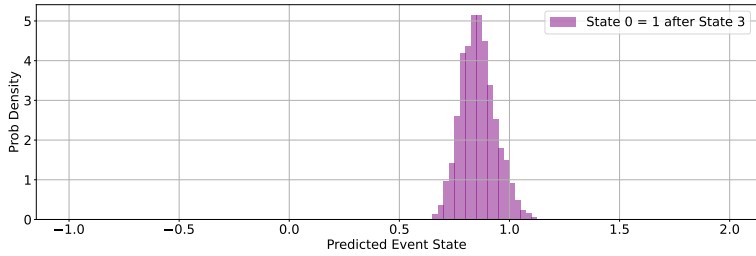

Figure 15: Prob density of predicted event state 0 after 3

In Figure 16, Figure 17 and Figure 18, we can observe the attention value distributions across different patches. The toy dataset has an input sequence length of 336, with a patch length of 16 and a stride of 16, while the event period is 80. As a result, the input sequence corresponds to 21 patches (tokens), with 4-5 patches containing event-related information (Event Patches).

From these figures, we can see that the attention distributions of event patches and non-event patches are nearly identical. In a Transformer-based model, this is highly unreasonable. Ideally, the model should focus on the last one or two event patches to predict the next event patch, meaning their attention scores should be significantly higher than those of other patches. However, our statistical analysis clearly demonstrates that the attention mechanism fails to capture meaningful contextual information.

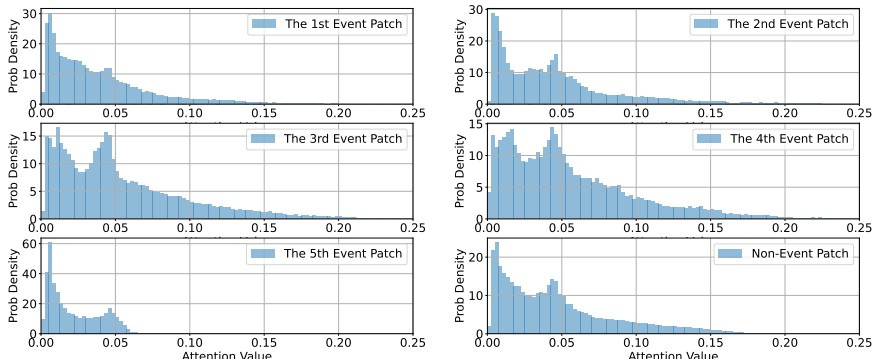

Figure 16: Block 0 attention value to patches in the input sequence

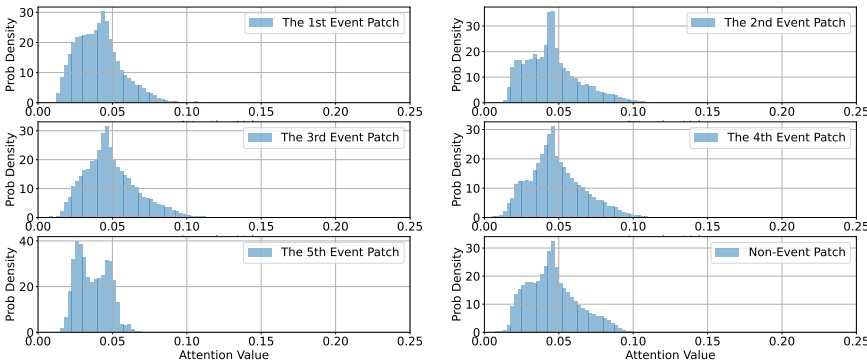

Figure 17: Block 1 attention value to patches in the input sequence

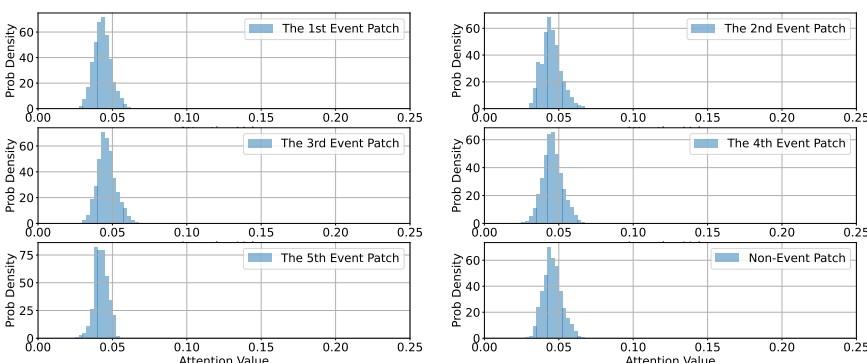

Figure 18: Block 2 attention value to patches in the input sequence

Additionally, the visualization of multi-head attention is overly complex, making it difficult to extract meaningful and convincing insights. We observed that, on the toy dataset, the performance difference between the 16-head attention model and the single-head attention model is minimal, with both models predicting S=2 as S=1. Therefore, for attention visualization, we only use the single-head attention model. In fact, for multi-head attention models, the attention distribution across most heads tends to resemble a uniform distribution, with little focus given to the event patch.

For PatchTST, we visualized the original model with no padding or with padding set to 1, and the experimental results were consistent with those presented in the main text. However, since the output depends on all tokens, the attention is not solely dominated by the event patch. We modified the model architecture by padding the 96 output points with 8 new tokens and then observed the attention of specific event-related tokens with respect to the input sequence.

### G.3 ATTENTION REPLACEMENT EXPERIMENT

We present the attention replacement experiment results for different models across various datasets. The findings align with our analysis in the main text. Due to differences in model implementations, the datasets, metrics, and attention module may vary across models. However, our primary focus is on comparing the performance of different attention types within the same scenario, making these variations negligible. The error bars in the following tables are the standard deviation and are obtained by taking different seeds during training. 'RAW', 'EYE', 'ZERO' and 'MEAN' in the tables below represents the types of attention.

Table 8: PatchTST attention replacement. Prediction length = 96.

| DATASET | METRIC | RAW | EYE | ZERO | MEAN |
|---|---|---|---|---|---|
| ETTH1 | MSE | $0.404 \pm 0.003$ | $0.381 \pm 0.000$ | $0.384 \pm 0.001$ | $0.392 \pm 0.001$ |
| ETTH1 | MAE | $0.418 \pm 0.001$ | $0.403 \pm 0.000$ | $0.405 \pm 0.000$ | $0.412 \pm 0.002$ |
| ETTH1 | MDA | $0.579 \pm 0.004$ | $0.588 \pm 0.002$ | $0.582 \pm 0.001$ | $0.585 \pm 0.001$ |
| ETTH2 | MSE | $0.306 \pm 0.003$ | $0.290 \pm 0.002$ | $0.291 \pm 0.001$ | $0.295 \pm 0.007$ |
| ETTH2 | MAE | $0.361 \pm 0.003$ | $0.350 \pm 0.002$ | $0.351 \pm 0.001$ | $0.355 \pm 0.005$ |
| ETTH2 | MDA | $0.412 \pm 0.002$ | $0.418 \pm 0.000$ | $0.417 \pm 0.001$ | $0.414 \pm 0.001$ |
| ETTM1 | MSE | $0.304 \pm 0.004$ | $0.294 \pm 0.002$ | $0.294 \pm 0.002$ | $0.300 \pm 0.001$ |
| ETTM1 | MAE | $0.358 \pm 0.003$ | $0.347 \pm 0.002$ | $0.348 \pm 0.002$ | $0.354 \pm 0.001$ |
| ETTM1 | MDA | $0.495 \pm 0.000$ | $0.500 \pm 0.000$ | $0.499 \pm 0.000$ | $0.498 \pm 0.000$ |
| ETTM2 | MSE | $0.174 \pm 0.003$ | $0.174 \pm 0.001$ | $0.174 \pm 0.000$ | $0.175 \pm 0.000$ |
| ETTM2 | MAE | $0.265 \pm 0.001$ | $0.264 \pm 0.000$ | $0.264 \pm 0.000$ | $0.265 \pm 0.000$ |
| ETTM2 | MDA | $0.322 \pm 0.001$ | $0.342 \pm 0.000$ | $0.342 \pm 0.000$ | $0.325 \pm 0.001$ |
| WEATHER | MSE | $0.162 \pm 0.005$ | $0.152 \pm 0.003$ | $0.152 \pm 0.003$ | $0.154 \pm 0.001$ |
| WEATHER | MAE | $0.211 \pm 0.005$ | $0.201 \pm 0.004$ | $0.201 \pm 0.002$ | $0.204 \pm 0.001$ |
| WEATHER | MDA | $0.424 \pm 0.001$ | $0.469 \pm 0.004$ | $0.468 \pm 0.002$ | $0.427 \pm 0.001$ |

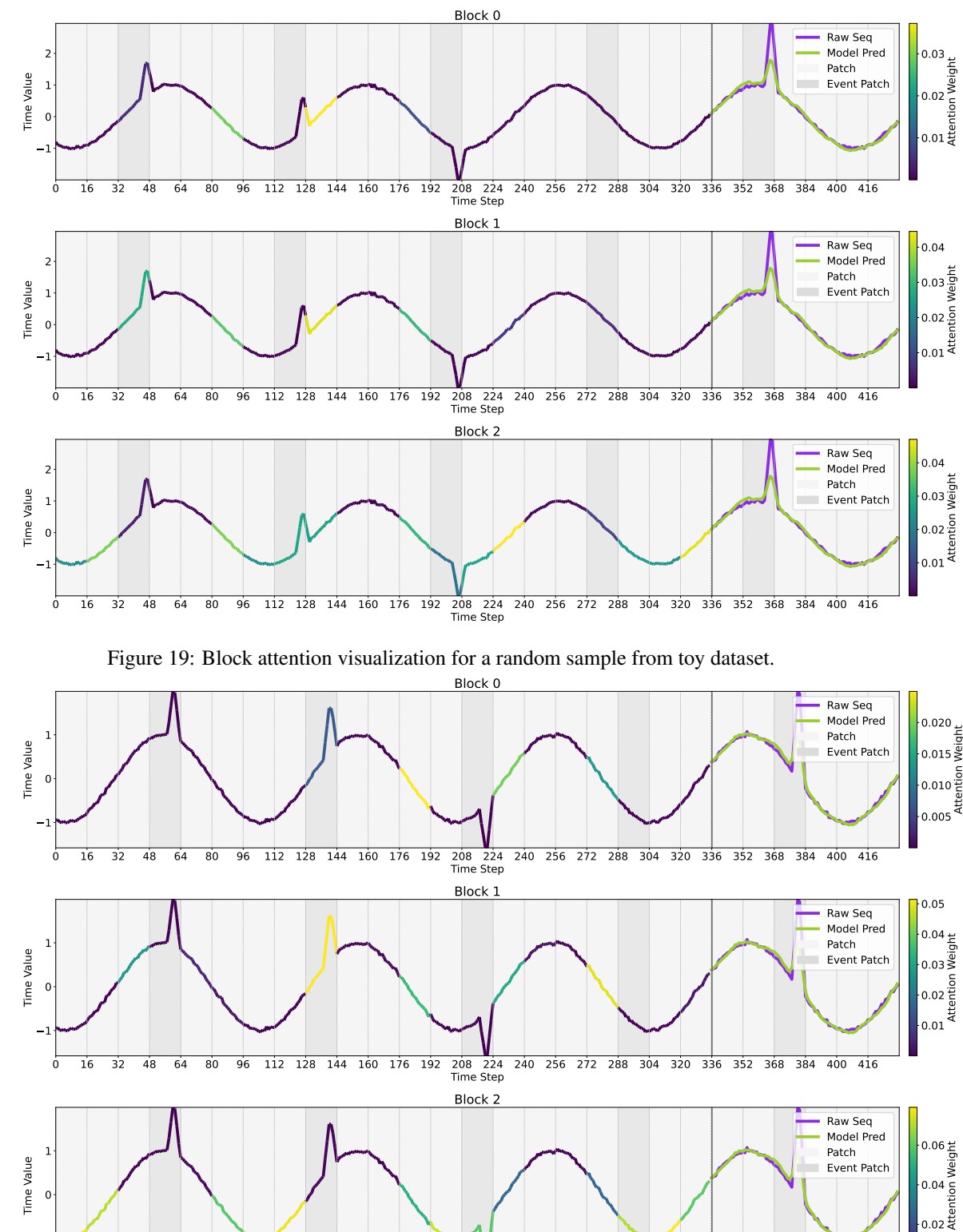

Figure 19: Block attention visualization for a random sample from toy dataset.

Figure 20: Block attention visualization for a random sample from toy dataset.

Table 9: PatchTST attention replacement. Prediction length = 192.

| DATASET | METRIC | RAW | EYE | ZERO | MEAN |
|---------|--------|-----|-----|------|------|
| ETTH1 | MSE | $0.451 \pm 0.007$ | $0.416 \pm 0.002$ | $0.419 \pm 0.002$ | $0.437 \pm 0.002$ |
| ETTH1 | MAE | $0.448 \pm 0.004$ | $0.426 \pm 0.002$ | $0.428 \pm 0.002$ | $0.443 \pm 0.001$ |
| ETTH1 | MDA | $0.571 \pm 0.001$ | $0.580 \pm 0.003$ | $0.573 \pm 0.003$ | $0.575 \pm 0.003$ |
| ETTH2 | MSE | $0.391 \pm 0.002$ | $0.356 \pm 0.002$ | $0.358 \pm 0.002$ | $0.361 \pm 0.002$ |
| ETTH2 | MAE | $0.412 \pm 0.001$ | $0.391 \pm 0.001$ | $0.392 \pm 0.001$ | $0.396 \pm 0.001$ |
| ETTH2 | MDA | $0.410 \pm 0.000$ | $0.416 \pm 0.001$ | $0.414 \pm 0.001$ | $0.410 \pm 0.002$ |
| ETTM1 | MSE | $0.345 \pm 0.000$ | $0.333 \pm 0.000$ | $0.333 \pm 0.000$ | $0.339 \pm 0.001$ |
| ETTM1 | MAE | $0.382 \pm 0.001$ | $0.371 \pm 0.000$ | $0.371 \pm 0.000$ | $0.377 \pm 0.000$ |
| ETTM1 | MDA | $0.497 \pm 0.000$ | $0.502 \pm 0.000$ | $0.500 \pm 0.000$ | $0.499 \pm 0.000$ |
| ETTM2 | MSE | $0.248 \pm 0.003$ | $0.237 \pm 0.002$ | $0.238 \pm 0.002$ | $0.236 \pm 0.000$ |
| ETTM2 | MAE | $0.315 \pm 0.000$ | $0.307 \pm 0.003$ | $0.307 \pm 0.004$ | $0.309 \pm 0.001$ |
| ETTM2 | MDA | $0.318 \pm 0.000$ | $0.338 \pm 0.001$ | $0.338 \pm 0.001$ | $0.320 \pm 0.000$ |
| WEATHER | MSE | $0.204 \pm 0.002$ | $0.194 \pm 0.000$ | $0.194 \pm 0.000$ | $0.199 \pm 0.001$ |
| WEATHER | MAE | $0.251 \pm 0.002$ | $0.242 \pm 0.001$ | $0.242 \pm 0.001$ | $0.245 \pm 0.000$ |
| WEATHER | MDA | $0.424 \pm 0.003$ | $0.468 \pm 0.001$ | $0.468 \pm 0.000$ | $0.429 \pm 0.001$ |

Table 10: PatchTST attention replacement. Prediction length = 336.

| DATASET | METRIC | RAW | EYE | ZERO | MEAN |
|---------|--------|-----|-----|------|------|
| ETTH1 | MSE | $0.471 \pm 0.002$ | $0.440 \pm 0.002$ | $0.445 \pm 0.000$ | $0.464 \pm 0.005$ |
| ETTH1 | MAE | $0.465 \pm 0.003$ | $0.444 \pm 0.003$ | $0.447 \pm 0.001$ | $0.462 \pm 0.001$ |
| ETTH1 | MDA | $0.563 \pm 0.001$ | $0.569 \pm 0.001$ | $0.560 \pm 0.001$ | $0.564 \pm 0.001$ |
| ETTH2 | MSE | $0.422 \pm 0.007$ | $0.381 \pm 0.001$ | $0.383 \pm 0.001$ | $0.384 \pm 0.002$ |
| ETTH2 | MAE | $0.435 \pm 0.005$ | $0.413 \pm 0.001$ | $0.414 \pm 0.001$ | $0.416 \pm 0.001$ |
| ETTH2 | MDA | $0.411 \pm 0.001$ | $0.418 \pm 0.000$ | $0.415 \pm 0.000$ | $0.412 \pm 0.002$ |
| ETTM1 | MSE | $0.379 \pm 0.000$ | $0.368 \pm 0.000$ | $0.368 \pm 0.000$ | $0.374 \pm 0.002$ |
| ETTM1 | MAE | $0.403 \pm 0.001$ | $0.391 \pm 0.000$ | $0.391 \pm 0.000$ | $0.398 \pm 0.001$ |
| ETTM1 | MDA | $0.496 \pm 0.001$ | $0.501 \pm 0.000$ | $0.500 \pm 0.000$ | $0.498 \pm 0.000$ |
| ETTM2 | MSE | $0.291 \pm 0.005$ | $0.286 \pm 0.001$ | $0.287 \pm 0.002$ | $0.292 \pm 0.002$ |
| ETTM2 | MAE | $0.343 \pm 0.003$ | $0.338 \pm 0.001$ | $0.338 \pm 0.001$ | $0.344 \pm 0.001$ |
| ETTM2 | MDA | $0.318 \pm 0.000$ | $0.336 \pm 0.000$ | $0.337 \pm 0.000$ | $0.319 \pm 0.002$ |
| WEATHER | MSE | $0.259 \pm 0.006$ | $0.246 \pm 0.000$ | $0.246 \pm 0.001$ | $0.251 \pm 0.002$ |
| WEATHER | MAE | $0.291 \pm 0.005$ | $0.283 \pm 0.000$ | $0.283 \pm 0.001$ | $0.285 \pm 0.001$ |
| WEATHER | MDA | $0.425 \pm 0.001$ | $0.469 \pm 0.001$ | $0.469 \pm 0.000$ | $0.431 \pm 0.005$ |

Table 11: PatchTST attention replacement. Prediction length = 720.

| DATASET | METRIC | RAW | EYE | ZERO | MEAN |
|---------|--------|-----|-----|------|------|
| ETTH1 | MSE | $0.559 \pm 0.069$ | $0.459 \pm 0.003$ | $0.467 \pm 0.005$ | $0.521 \pm 0.016$ |
| ETTH1 | MAE | $0.534 \pm 0.039$ | $0.476 \pm 0.002$ | $0.480 \pm 0.004$ | $0.514 \pm 0.010$ |
| ETTH1 | MDA | $0.553 \pm 0.001$ | $0.559 \pm 0.000$ | $0.554 \pm 0.000$ | $0.557 \pm 0.000$ |
| ETTH2 | MSE | $0.416 \pm 0.001$ | $0.410 \pm 0.001$ | $0.411 \pm 0.001$ | $0.419 \pm 0.003$ |
| ETTH2 | MAE | $0.443 \pm 0.002$ | $0.440 \pm 0.001$ | $0.441 \pm 0.001$ | $0.446 \pm 0.002$ |
| ETTH2 | MDA | $0.411 \pm 0.001$ | $0.417 \pm 0.000$ | $0.414 \pm 0.000$ | $0.412 \pm 0.002$ |
| ETTM1 | MSE | $0.442 \pm 0.006$ | $0.427 \pm 0.002$ | $0.426 \pm 0.002$ | $0.435 \pm 0.000$ |
| ETTM1 | MAE | $0.438 \pm 0.002$ | $0.425 \pm 0.002$ | $0.424 \pm 0.001$ | $0.433 \pm 0.001$ |
| ETTM1 | MDA | $0.492 \pm 0.004$ | $0.498 \pm 0.001$ | $0.497 \pm 0.002$ | $0.497 \pm 0.000$ |
| ETTM2 | MSE | $0.377 \pm 0.004$ | $0.369 \pm 0.002$ | $0.369 \pm 0.002$ | $0.375 \pm 0.002$ |
| ETTM2 | MAE | $0.395 \pm 0.002$ | $0.390 \pm 0.000$ | $0.390 \pm 0.000$ | $0.395 \pm 0.001$ |
| ETTM2 | MDA | $0.320 \pm 0.003$ | $0.336 \pm 0.000$ | $0.336 \pm 0.001$ | $0.321 \pm 0.001$ |
| WEATHER | MSE | $0.326 \pm 0.004$ | $0.322 \pm 0.001$ | $0.323 \pm 0.000$ | $0.323 \pm 0.003$ |
| WEATHER | MAE | $0.338 \pm 0.001$ | $0.337 \pm 0.000$ | $0.337 \pm 0.000$ | $0.334 \pm 0.001$ |
| WEATHER | MDA | $0.427 \pm 0.002$ | $0.472 \pm 0.000$ | $0.472 \pm 0.000$ | $0.434 \pm 0.006$ |

Table 12: iTransformer attention replacement. Prediction length = 96.

| DATASET | METRIC | RAW | EYE | ZERO | MEAN |
|---|---|---|---|---|---|
| ETTH1 | MSE | $0.404 \pm 0.003$ | $0.381 \pm 0.000$ | $0.384 \pm 0.001$ | $0.392 \pm 0.001$ |
| ETTH1 | MAE | $0.418 \pm 0.001$ | $0.403 \pm 0.000$ | $0.405 \pm 0.000$ | $0.412 \pm 0.002$ |
| ETTH1 | MDA | $0.579 \pm 0.004$ | $0.588 \pm 0.002$ | $0.582 \pm 0.001$ | $0.585 \pm 0.001$ |
| ETTH2 | MSE | $0.306 \pm 0.003$ | $0.290 \pm 0.002$ | $0.291 \pm 0.001$ | $0.295 \pm 0.007$ |
| ETTH2 | MAE | $0.361 \pm 0.003$ | $0.350 \pm 0.002$ | $0.351 \pm 0.001$ | $0.355 \pm 0.005$ |
| ETTH2 | MDA | $0.412 \pm 0.002$ | $0.418 \pm 0.000$ | $0.417 \pm 0.001$ | $0.414 \pm 0.001$ |
| ETTM1 | MSE | $0.304 \pm 0.004$ | $0.294 \pm 0.002$ | $0.294 \pm 0.002$ | $0.300 \pm 0.001$ |
| ETTM1 | MAE | $0.358 \pm 0.003$ | $0.347 \pm 0.002$ | $0.348 \pm 0.002$ | $0.354 \pm 0.001$ |
| ETTM1 | MDA | $0.495 \pm 0.000$ | $0.500 \pm 0.000$ | $0.499 \pm 0.000$ | $0.498 \pm 0.000$ |
| ETTM2 | MSE | $0.174 \pm 0.003$ | $0.174 \pm 0.001$ | $0.174 \pm 0.000$ | $0.175 \pm 0.000$ |
| ETTM2 | MAE | $0.265 \pm 0.001$ | $0.264 \pm 0.000$ | $0.264 \pm 0.000$ | $0.265 \pm 0.000$ |
| ETTM2 | MDA | $0.322 \pm 0.001$ | $0.342 \pm 0.000$ | $0.342 \pm 0.000$ | $0.325 \pm 0.001$ |
| WEATHER | MSE | $0.162 \pm 0.005$ | $0.152 \pm 0.003$ | $0.152 \pm 0.003$ | $0.154 \pm 0.001$ |
| WEATHER | MAE | $0.211 \pm 0.005$ | $0.201 \pm 0.004$ | $0.201 \pm 0.002$ | $0.204 \pm 0.001$ |
| WEATHER | MDA | $0.424 \pm 0.001$ | $0.469 \pm 0.004$ | $0.468 \pm 0.002$ | $0.427 \pm 0.001$ |

Table 13: iTransformer attention replacement. Prediction length = 192.

| DATASET | METRIC | RAW | EYE | ZERO | MEAN |
|---|---|---|---|---|---|
| ETTH1 | MSE | $0.451 \pm 0.007$ | $0.416 \pm 0.002$ | $0.419 \pm 0.002$ | $0.437 \pm 0.002$ |
| ETTH1 | MAE | $0.448 \pm 0.004$ | $0.426 \pm 0.002$ | $0.428 \pm 0.002$ | $0.443 \pm 0.001$ |
| ETTH1 | MDA | $0.571 \pm 0.001$ | $0.580 \pm 0.003$ | $0.573 \pm 0.003$ | $0.575 \pm 0.003$ |
| ETTH2 | MSE | $0.391 \pm 0.002$ | $0.356 \pm 0.002$ | $0.358 \pm 0.002$ | $0.361 \pm 0.002$ |
| ETTH2 | MAE | $0.412 \pm 0.001$ | $0.391 \pm 0.001$ | $0.392 \pm 0.001$ | $0.396 \pm 0.001$ |
| ETTH2 | MDA | $0.410 \pm 0.000$ | $0.416 \pm 0.001$ | $0.414 \pm 0.001$ | $0.410 \pm 0.002$ |
| ETTM1 | MSE | $0.345 \pm 0.000$ | $0.333 \pm 0.000$ | $0.333 \pm 0.000$ | $0.339 \pm 0.001$ |
| ETTM1 | MAE | $0.382 \pm 0.001$ | $0.371 \pm 0.000$ | $0.371 \pm 0.000$ | $0.377 \pm 0.000$ |
| ETTM1 | MDA | $0.497 \pm 0.000$ | $0.502 \pm 0.000$ | $0.500 \pm 0.000$ | $0.499 \pm 0.000$ |
| ETTM2 | MSE | $0.248 \pm 0.003$ | $0.237 \pm 0.002$ | $0.238 \pm 0.002$ | $0.236 \pm 0.000$ |
| ETTM2 | MAE | $0.315 \pm 0.000$ | $0.307 \pm 0.003$ | $0.307 \pm 0.004$ | $0.309 \pm 0.001$ |
| ETTM2 | MDA | $0.318 \pm 0.000$ | $0.338 \pm 0.001$ | $0.338 \pm 0.001$ | $0.320 \pm 0.000$ |
| WEATHER | MSE | $0.204 \pm 0.002$ | $0.194 \pm 0.000$ | $0.194 \pm 0.000$ | $0.199 \pm 0.001$ |
| WEATHER | MAE | $0.251 \pm 0.002$ | $0.242 \pm 0.001$ | $0.242 \pm 0.001$ | $0.245 \pm 0.000$ |
| WEATHER | MDA | $0.424 \pm 0.003$ | $0.468 \pm 0.001$ | $0.468 \pm 0.000$ | $0.429 \pm 0.001$ |

Table 14: iTransformer attention replacement. Prediction length = 336.

| DATASET | METRIC | RAW | EYE | ZERO | MEAN |
|---|---|---|---|---|---|
| ETTH1 | MSE | $0.471 \pm 0.002$ | $0.440 \pm 0.002$ | $0.445 \pm 0.000$ | $0.464 \pm 0.005$ |
| ETTH1 | MAE | $0.465 \pm 0.003$ | $0.444 \pm 0.003$ | $0.447 \pm 0.001$ | $0.462 \pm 0.001$ |
| ETTH1 | MDA | $0.563 \pm 0.001$ | $0.569 \pm 0.001$ | $0.560 \pm 0.001$ | $0.564 \pm 0.001$ |
| ETTH2 | MSE | $0.422 \pm 0.007$ | $0.381 \pm 0.001$ | $0.383 \pm 0.001$ | $0.384 \pm 0.002$ |
| ETTH2 | MAE | $0.435 \pm 0.005$ | $0.413 \pm 0.001$ | $0.414 \pm 0.001$ | $0.416 \pm 0.001$ |
| ETTH2 | MDA | $0.411 \pm 0.001$ | $0.418 \pm 0.000$ | $0.415 \pm 0.000$ | $0.412 \pm 0.002$ |
| ETTM1 | MSE | $0.379 \pm 0.000$ | $0.368 \pm 0.000$ | $0.368 \pm 0.000$ | $0.374 \pm 0.002$ |
| ETTM1 | MAE | $0.403 \pm 0.001$ | $0.391 \pm 0.000$ | $0.391 \pm 0.000$ | $0.398 \pm 0.001$ |
| ETTM1 | MDA | $0.496 \pm 0.001$ | $0.501 \pm 0.000$ | $0.500 \pm 0.000$ | $0.498 \pm 0.000$ |
| ETTM2 | MSE | $0.291 \pm 0.005$ | $0.286 \pm 0.001$ | $0.287 \pm 0.002$ | $0.292 \pm 0.002$ |
| ETTM2 | MAE | $0.343 \pm 0.003$ | $0.338 \pm 0.001$ | $0.338 \pm 0.001$ | $0.344 \pm 0.001$ |
| ETTM2 | MDA | $0.318 \pm 0.000$ | $0.336 \pm 0.000$ | $0.337 \pm 0.000$ | $0.319 \pm 0.002$ |
| WEATHER | MSE | $0.259 \pm 0.006$ | $0.246 \pm 0.000$ | $0.246 \pm 0.001$ | $0.251 \pm 0.002$ |
| WEATHER | MAE | $0.291 \pm 0.005$ | $0.283 \pm 0.000$ | $0.283 \pm 0.001$ | $0.285 \pm 0.001$ |
| WEATHER | MDA | $0.425 \pm 0.001$ | $0.469 \pm 0.001$ | $0.469 \pm 0.000$ | $0.431 \pm 0.005$ |

Table 15: iTransformer attention replacement. Prediction length = 720.

| DATASET | METRIC | RAW | EYE | ZERO | MEAN |
|---|---|---|---|---|---|
| ETTH1 | MSE | $0.559 \pm 0.069$ | $0.459 \pm 0.003$ | $0.467 \pm 0.005$ | $0.521 \pm 0.016$ |
| ETTH1 | MAE | $0.534 \pm 0.039$ | $0.476 \pm 0.002$ | $0.480 \pm 0.004$ | $0.514 \pm 0.010$ |
| ETTH1 | MDA | $0.553 \pm 0.001$ | $0.559 \pm 0.000$ | $0.554 \pm 0.000$ | $0.557 \pm 0.000$ |
| ETTH2 | MSE | $0.416 \pm 0.001$ | $0.410 \pm 0.001$ | $0.411 \pm 0.001$ | $0.419 \pm 0.003$ |
| ETTH2 | MAE | $0.443 \pm 0.002$ | $0.440 \pm 0.001$ | $0.441 \pm 0.001$ | $0.446 \pm 0.002$ |
| ETTH2 | MDA | $0.411 \pm 0.001$ | $0.417 \pm 0.000$ | $0.414 \pm 0.000$ | $0.412 \pm 0.000$ |
| ETTM1 | MSE | $0.442 \pm 0.006$ | $0.427 \pm 0.002$ | $0.426 \pm 0.002$ | $0.435 \pm 0.000$ |
| ETTM1 | MAE | $0.438 \pm 0.002$ | $0.425 \pm 0.002$ | $0.424 \pm 0.001$ | $0.433 \pm 0.001$ |
| ETTM1 | MDA | $0.492 \pm 0.004$ | $0.498 \pm 0.001$ | $0.497 \pm 0.002$ | $0.497 \pm 0.000$ |
| ETTM2 | MSE | $0.377 \pm 0.004$ | $0.369 \pm 0.002$ | $0.369 \pm 0.002$ | $0.375 \pm 0.002$ |
| ETTM2 | MAE | $0.395 \pm 0.002$ | $0.390 \pm 0.000$ | $0.390 \pm 0.000$ | $0.395 \pm 0.001$ |
| ETTM2 | MDA | $0.320 \pm 0.003$ | $0.336 \pm 0.000$ | $0.336 \pm 0.001$ | $0.321 \pm 0.001$ |
| WEATHER | MSE | $0.326 \pm 0.004$ | $0.322 \pm 0.001$ | $0.323 \pm 0.000$ | $0.323 \pm 0.003$ |
| WEATHER | MAE | $0.338 \pm 0.001$ | $0.337 \pm 0.000$ | $0.337 \pm 0.000$ | $0.334 \pm 0.001$ |
| WEATHER | MDA | $0.427 \pm 0.002$ | $0.472 \pm 0.000$ | $0.472 \pm 0.000$ | $0.434 \pm 0.006$ |

Table 16: Pathformer attention replacement. Prediction length = 96.

| DATASET | METRIC | RAW | EYE | ZERO |
|---|---|---|---|---|
| ETTH1 | MSE | $0.384 \pm 0.001$ | $0.383 \pm 0.001$ | $0.384 \pm 0.002$ |
| ETTH1 | MAE | $0.390 \pm 0.002$ | $0.388 \pm 0.001$ | $0.389 \pm 0.002$ |
| ETTH1 | MDA | $0.600 \pm 0.003$ | $0.600 \pm 0.005$ | $0.600 \pm 0.004$ |
| ETTH2 | MSE | $0.294 \pm 0.003$ | $0.283 \pm 0.001$ | $0.285 \pm 0.000$ |
| ETTH2 | MAE | $0.339 \pm 0.002$ | $0.333 \pm 0.001$ | $0.334 \pm 0.000$ |
| ETTH2 | MDA | $0.434 \pm 0.000$ | $0.433 \pm 0.000$ | $0.436 \pm 0.003$ |
| ETTM1 | MSE | $0.315 \pm 0.003$ | $0.314 \pm 0.003$ | $0.314 \pm 0.003$ |
| ETTM1 | MAE | $0.345 \pm 0.001$ | $0.344 \pm 0.002$ | $0.343 \pm 0.001$ |
| ETTM1 | MDA | $0.498 \pm 0.000$ | $0.498 \pm 0.000$ | $0.499 \pm 0.000$ |
| ETTM2 | MSE | $0.168 \pm 0.001$ | $0.174 \pm 0.000$ | $0.173 \pm 0.002$ |
| ETTM2 | MAE | $0.248 \pm 0.001$ | $0.252 \pm 0.000$ | $0.252 \pm 0.002$ |
| ETTM2 | MDA | $0.328 \pm 0.004$ | $0.330 \pm 0.001$ | $0.335 \pm 0.002$ |

Table 17: Pathformer attention replacement. Prediction length = 192.

| DATASET | METRIC | RAW | EYE | ZERO |
|---|---|---|---|---|
| ETTH1 | MSE | $0.442 \pm 0.000$ | $0.442 \pm 0.004$ | $0.441 \pm 0.005$ |
| ETTH1 | MAE | $0.419 \pm 0.001$ | $0.418 \pm 0.001$ | $0.418 \pm 0.001$ |
| ETTH1 | MDA | $0.593 \pm 0.002$ | $0.595 \pm 0.005$ | $0.595 \pm 0.004$ |
| ETTH2 | MSE | $0.367 \pm 0.006$ | $0.358 \pm 0.003$ | $0.361 \pm 0.001$ |
| ETTH2 | MAE | $0.386 \pm 0.004$ | $0.380 \pm 0.003$ | $0.382 \pm 0.001$ |
| ETTH2 | MDA | $0.432 \pm 0.006$ | $0.435 \pm 0.006$ | $0.434 \pm 0.004$ |
| ETTM1 | MSE | $0.364 \pm 0.004$ | $0.365 \pm 0.002$ | $0.366 \pm 0.005$ |
| ETTM1 | MAE | $0.368 \pm 0.002$ | $0.368 \pm 0.002$ | $0.368 \pm 0.002$ |
| ETTM1 | MDA | $0.500 \pm 0.001$ | $0.501 \pm 0.001$ | $0.501 \pm 0.002$ |
| ETTM2 | MSE | $0.233 \pm 0.001$ | $0.236 \pm 0.001$ | $0.237 \pm 0.000$ |
| ETTM2 | MAE | $0.292 \pm 0.002$ | $0.293 \pm 0.000$ | $0.294 \pm 0.001$ |
| ETTM2 | MDA | $0.326 \pm 0.004$ | $0.329 \pm 0.001$ | $0.333 \pm 0.004$ |

Table 18: Pathformer attention replacement. Prediction length = 336.

| DATASET | METRIC | RAW | EYE | ZERO |
|---------|--------|-----|-----|------|
| ETTH1 | MSE | $0.461 \pm 0.007$ | $0.455 \pm 0.007$ | $0.455 \pm 0.004$ |
| ETTH1 | MAE | $0.431 \pm 0.003$ | $0.425 \pm 0.003$ | $0.425 \pm 0.001$ |
| ETTH1 | MDA | $0.579 \pm 0.013$ | $0.583 \pm 0.006$ | $0.584 \pm 0.006$ |
| ETTH2 | MSE | $0.382 \pm 0.017$ | $0.345 \pm 0.001$ | $0.347 \pm 0.003$ |
| ETTH2 | MAE | $0.406 \pm 0.012$ | $0.381 \pm 0.001$ | $0.383 \pm 0.002$ |
| ETTH2 | MDA | $0.419 \pm 0.018$ | $0.429 \pm 0.000$ | $0.427 \pm 0.009$ |
| ETTM1 | MSE | $0.386 \pm 0.000$ | $0.386 \pm 0.002$ | $0.390 \pm 0.006$ |
| ETTM1 | MAE | $0.389 \pm 0.001$ | $0.388 \pm 0.001$ | $0.390 \pm 0.002$ |
| ETTM1 | MDA | $0.498 \pm 0.001$ | $0.498 \pm 0.001$ | $0.499 \pm 0.002$ |
| ETTM2 | MSE | $0.292 \pm 0.000$ | $0.298 \pm 0.001$ | $0.297 \pm 0.002$ |
| ETTM2 | MAE | $0.329 \pm 0.001$ | $0.332 \pm 0.000$ | $0.333 \pm 0.000$ |
| ETTM2 | MDA | $0.328 \pm 0.001$ | $0.329 \pm 0.000$ | $0.334 \pm 0.001$ |

Table 19: Pathformer attention replacement. Prediction length = 720.

| DATASET | METRIC | RAW | EYE | ZERO |
|---------|--------|-----|-----|------|
| ETTH1 | MSE | $0.492 \pm 0.008$ | $0.484 \pm 0.004$ | $0.485 \pm 0.004$ |
| ETTH1 | MAE | $0.464 \pm 0.003$ | $0.455 \pm 0.003$ | $0.457 \pm 0.004$ |
| ETTH1 | MDA | $0.569 \pm 0.000$ | $0.572 \pm 0.003$ | $0.572 \pm 0.002$ |
| ETTH2 | MSE | $0.399 \pm 0.014$ | $0.398 \pm 0.008$ | $0.415 \pm 0.002$ |
| ETTH2 | MAE | $0.423 \pm 0.007$ | $0.423 \pm 0.004$ | $0.434 \pm 0.002$ |
| ETTH2 | MDA | $0.424 \pm 0.003$ | $0.429 \pm 0.000$ | $0.429 \pm 0.001$ |
| ETTM1 | MSE | $0.456 \pm 0.008$ | $0.456 \pm 0.004$ | $0.443 \pm 0.035$ |
| ETTM1 | MAE | $0.428 \pm 0.002$ | $0.426 \pm 0.002$ | $0.442 \pm 0.032$ |
| ETTM1 | MDA | $0.493 \pm 0.002$ | $0.494 \pm 0.002$ | $0.495 \pm 0.003$ |
| ETTM2 | MSE | $0.394 \pm 0.018$ | $0.391 \pm 0.004$ | $0.389 \pm 0.003$ |
| ETTM2 | MAE | $0.390 \pm 0.008$ | $0.388 \pm 0.002$ | $0.389 \pm 0.001$ |
| ETTM2 | MDA | $0.320 \pm 0.007$ | $0.327 \pm 0.001$ | $0.332 \pm 0.002$ |

Table 20: Lag-llama attention replacement(CRPS).

| Dataset | Airpassengers | ECL | Exchange | Hospital | Pedestrian | Saugeenday | Solar | Taxi | Traffic | Weather |
|---------|---------------|-----|----------|----------|------------|------------|-------|------|---------|---------|
| **RAW** | 0.118 | 0.050 | **0.015** | 0.120 | 0.216 | 0.605 | **0.381** | **0.306** | 0.114 | 0.149 |
| **MEAN** | **0.069** | 0.068 | 0.019 | 0.087 | **0.182** | 0.607 | 0.489 | 0.322 | 0.114 | 0.150 |
| **EYE** | 0.157 | **0.044** | 0.115 | **0.065** | 0.202 | **0.547** | 0.404 | 0.311 | 0.114 | 0.153 |
| **ZERO** | 0.107 | 0.051 | 0.059 | 0.072 | 0.231 | 0.565 | 0.387 | 0.318 | 0.120 | **0.147** |

Table 21: SAMformer attention replacement(RMSE).

| DATASET | ETTH1 | ETTH2 | ETTM1 | ECL | WEATHER |
|---------|-------|-------|-------|-----|---------|
| **RAW** | 0.645 | 0.590 | 0.728 | **0.409** | **1.118** |
| **MEAN** | **0.642** | 0.587 | **0.713** | 0.412 | 1.133 |
| **EYE** | 0.649 | 0.588 | 0.725 | 0.410 | 1.124 |
| **ZERO** | 0.647 | **0.584** | 0.736 | 0.411 | 1.141 |

Table 22: Timer attention replacement(MSE).

| DATASET | ECL | ETTH1 | TRAFFIC | WEATHER |
|---------|------|-------|---------|---------|
| **RAW** | **0.133** | 0.368 | **0.350** | **0.166** |
| **MEAN** | 0.135 | **0.365** | 0.358 | 0.172 |
| **EYE** | 0.155 | 0.382 | 0.397 | 0.174 |
| **ZERO** | 0.154 | 0.393 | 0.394 | 0.171 |

Table 23: Moirai attention replacement(NRMSE).

| DATASET | **RAW** | **FIX** |
|---------|---------|---------|
| ETTH1 | **0.512** | 0.523 |

### G.4 POSITIONAL ENCODING ZEROING OUT EXPERIMENT

We present the results of the Positional Encoding Zeroing Out Experiment in Table 24 and Table 25. The findings are consistent with our analysis in the main text.

Table 24: iTransformer fix embedding replacement.

| DATASET | ECL | ETTH1 | ETTH2 | ETTM1 | ETTM2 | EXCHANGE | TRAFFIC | WEATHER |
|---------|------|-------|-------|-------|-------|----------|---------|---------|
| **RAW EMB** | 0.153 | 0.385 | 0.300 | 0.345 | 0.187 | 0.092 | 0.400 | 0.177 |
| **FIXED EMB** | 0.154 | 0.388 | 0.302 | 0.343 | 0.187 | 0.090 | 0.399 | 0.177 |

Table 25: PatchTST fixed embedding experiment.

| METRIC | ECL | TRAFFIC | WEATHER | ETTH1 | ETTH2 | ETTM1 | ETTM2 |
|--------|------|---------|---------|-------|-------|-------|-------|
| **RAW EMB** | 0.130 | 0.371 | 0.153 | 0.383 | 0.277 | 0.285 | 0.163 |
| **FIXED EMB** | 0.130 | 0.372 | 0.152 | 0.382 | 0.277 | 0.286 | 0.163 |

### G.5 VIT AS AN EXCEPTION

Our ViT model is trained directly on CIFAR-10, with a total of 10 classes. The patch size is 4, the hidden size is 256, and the model consists of 8 blocks, each with 8 heads. We did not pre-train it on ImageNet.

We present the experimental results of PatchTST and iTransformer with varying numbers of blocks. It can be observed that the performance improvement with an increasing number of blocks is not significant and, in some cases, even leads to overfitting, such as in the case of iTransformer on the ETT dataset.

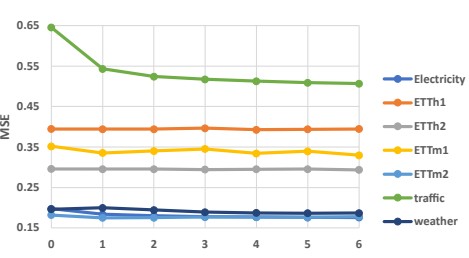

Figure 21: MSE performance of PatchTST with varying block numbers on different datasets.

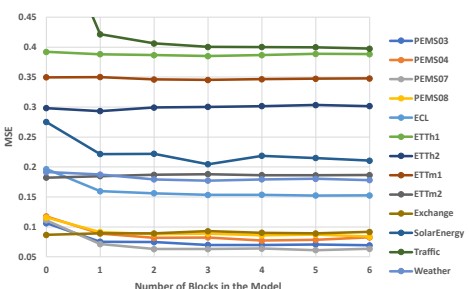

Figure 22: MSE performance of iTransformer with varying block numbers on different datasets.

## H  MATHEMATICAL PROOF

To demonstrate that linear embedding is not an effective method, we provide a simple constructive proof. The core of this proof is that in a temporal scenario, a linear layer cannot satisfy the desired properties of a latent space. Decomposing this problem, our proof needs to include the following elements:

- **Property A**: The temporal data satisfies Property A.

- **Property B**: The latent space needs to satisfy Property B.

- After passing through a linear layer, Property A conflicts with Property B.

To ensure broad applicability, we want Property B to be as weak as possible so that they are widely acceptable. We consider the following candidates (note that these property candidates are used for discussion purposes; their naming and descriptions are not formal):

- **Clustering Property**: Similar input data should be close in the latent space. This property can be analyzed by measuring latent space distances, such as Euclidean distance or cosine similarity.
  Consids: Clustering Property is widely used in latent space modeling across various neural network domains. Many studies and tools employ contrastive learning to enforce Clustering Property and achieve strong performance, including in the temporal domain. Since we are unsure whether Clustering Property is a formally accepted and rigorous term, we did not explicitly mention it in the paper. Instead, we used contrastive learning and Definition H.1 to describe this property.

- **Disentanglement**
  Consids: This property is mainly applicable to VAE models, where different channels in the latent space encode different feature factors. However, enforcing Disentanglement would require the decomposition of an N-dimensional space into N orthogonal feature factors, which is not a requirement in current Transformer models. According to the Johnson-Lindenstrauss lemma, a sufficiently large latent space can accommodate an exponential number of orthogonal vectors, allowing the latent space to encode a vast number of semantic directions.

- **Distributional Properties** (Used in VAE, not for transformers)

- **Locally Linear** (Difficult to model)

In summary, we adopt the Clustering Property required by contrastive learning as our Property B, since other properties do not meet the required criteria. Contrastive learning is a widely used training technique, and the Clustering Property is also a highly desirable characteristic. Many influential academic papers have employed contrastive learning to obtain better latent space representations (i.e., to obtain the Clustering Property)(Pöppelbaum et al., 2022; Lee et al., 2024a;b; Luo et al., 2023). Hence, We think adopting Clustering Property as Property B is acceptable.

To obtain Property A, given that most existing models default to using a linear layer as the embedding layer, we only need to find a dataset (the definition of the dataset itself is Property A.) that satisfies Proof C to challenge the effectiveness of linear embedding layers. Among various datasets, trigonometric and trapezoidal wave signals are the simplest and most commonly encountered signal types. Their spectral distributions differ significantly after Fourier transformation, making them distinct signal types. In fact, trigonometric and trapezoidal wave signals are among the most fundamental primitive signal types in classical signal processing and are so prevalent in industry and daily life that if a linear embedding layer fails to distinguish them effectively in the latent space, we can justifiably question the validity of linear embedding layers, even if our proof only applies to these two signal types and lacks broader generalizability.

For sets $\mathcal{X}$ and $\mathcal{Y}$, which are collections of two different types of signals in signal processing, a robust representation learning mapping $\phi$ will minimize the intra-set distances while maximizing the inter-set distances.

**Definition H.1** (Clustering Property for Representation Learning, and the Optimization Objective of Contrastive Learning). Let $\mathcal{X}$ and $\mathcal{Y}$ denote two disjoint sets representing different semantic classes. A representation mapping $\phi$ is said to satisfy the *Clustering Property* if

$$\begin{aligned}
\|\phi(\mathbf{x}_1) - \phi(\mathbf{x}_2)\|_2 &< \|\phi(\mathbf{x}_1) - \phi(\mathbf{y}_1)\|_2 \\
\|\phi(\mathbf{x}_1) - \phi(\mathbf{y}_1)\|_2 &> \|\phi(\mathbf{y}_1) - \phi(\mathbf{y}_2)\|_2
\end{aligned} \tag{8}$$

for any $\mathbf{x}_1, \mathbf{x}_2 \in \mathcal{X}$ and $\mathbf{y}_1, \mathbf{y}_2 \in \mathcal{Y}$.

We propose Theorem H.1, which states that the linear embedding cannot satisfy the conditions of Definition H.1 in certain cases.

**Theorem H.1.** *There exist two sets $\mathcal{X}$ and $\mathcal{Y}$ such that for any linear transformation $\mathbf{M} \in \mathbb{R}^{d \times n}$, there exist $\mathbf{x}_i, \mathbf{x}_j \in \mathcal{X}$ and $\mathbf{y}_k \in \mathcal{Y}$ for which*

$$\|\mathbf{M}\mathbf{x}_i - \mathbf{M}\mathbf{x}_j\|_2 > \|\mathbf{M}\mathbf{x}_i - \mathbf{M}\mathbf{y}_k\|_2.$$

*Thus, no linear embedding can satisfy Definition H.1 simultaneously for these two sets.*

*Proof.* We construct a set $\mathcal{X}$ of trigonometric signals and a set $\mathcal{Y}$ of trapezoidal wave signals. In most signal processing scenarios, these two sets represent different classes due to the extremely different spectral distributions. It is easy to observe that the n-1 vectors in $\mathcal{X}$ shown in the following are mutually orthogonal. Note that for the expressions below, we have $i = 1, 2, \ldots, n$. Suppose that $n = 2s$, we have

$$\mathcal{X}' = \left\{ k = 1, \ldots, s \mid \left( \cos\left(2\pi k \frac{1}{n} + \varphi_k\right), \ldots, \cos\left(2\pi k \frac{i}{n} + \varphi_k\right), \ldots, \cos\left(2\pi k \frac{n}{n} + \varphi_k\right) \right)^\top, \varphi_k \in [-\pi, \pi] \right\}$$

$$\cup \left\{ k = 1, \ldots, s-1 \mid \left( \sin\left(2\pi k \frac{1}{n} + \psi_k\right), \ldots, \sin\left(2\pi k \frac{i}{n} + \psi_k\right), \ldots, \sin\left(2\pi k \frac{n}{n} + \psi_k\right) \right)^\top, \psi_k = \varphi_k + m\pi, m \in \mathbb{Z} \right\}$$

$$\tag{9}$$

$$\mathcal{X} = \left\{ \frac{\mathbf{x}'}{2\|\mathbf{x}'\|} \mid \mathbf{x}' \in \mathcal{X}' \right\} \tag{10}$$

$$\mathcal{Y} = \left\{ \frac{1}{4\|\mathbf{x}'\|} \times \text{sgn}(\mathbf{x}') \mid \mathbf{x}' \in \mathcal{X}' \right\} \tag{11}$$

Where for a scalar $x$, $\text{sgn}(x)$ is defined as:

$$\text{sgn}(x) = \begin{cases} 1, & \text{if } x > 0 \\ 0, & \text{if } x = 0 \\ -1, & \text{if } x < 0 \end{cases} \tag{12}$$

For a vector $\mathbf{x}$, $\text{sgn}(\mathbf{x})$ is obtained by applying $\text{sgn}(x_k)$ element-wise.

Let $\mathbf{x}_i \in \mathcal{X}$ and $\mathbf{y}_i \in \mathcal{Y}$, where $\mathcal{X}$ and $\mathcal{Y}$ represent sets of trigonometric and trapezoidal wave signals, respectively. Clearly, for the unit vector $\mathbf{v} = \frac{1}{\sqrt{n}}\mathbf{1}_n$, the pair $\{\mathcal{X}, \mathbf{v}\}$ forms an orthogonal basis. Given these vectors, we define $\mathbf{a}_i = \mathbf{y}_i - \mathbf{x}_i$ and $\mathbf{b}_i = \mathbf{x}_i - (-\mathbf{x}_i) = 2\mathbf{x}_i$. Note that $-\mathbf{x}_i$ and $\mathbf{x}_i$ belong to the same class of signals. Clearly, $|\mathbf{b}_i|^2 = 1$, and the vectors $\mathbf{b}_i$ are pairwise orthogonal.

For any index $i$, we have

$$\begin{aligned}
\|\mathbf{a}_i\| - \|\mathbf{b}_i\| &= \|\mathbf{y}_i - \mathbf{x}_i\| - \|2\mathbf{x}_i\| \\
&= \sqrt{\sum_k (y_{ki} - x_{ki})^2} - 2\|\mathbf{x}_i\|, \; y_{ki} \text{ and } x'_{ki} \text{ have the same sign.} \\
&= \frac{1}{\|\mathbf{x}'_i\|}\sqrt{\sum_{k=1}^{n}\left(\frac{1}{4} - \frac{|x'_{ki}|}{2}\right)^2} - 1, \; 0 \le |x'_{ki}| \le 1 \\
&< \frac{1}{\|\mathbf{x}'_i\|}\sqrt{\sum_{k=1}^{n}\left(\frac{1}{4}\right)^2} - 1, \|\mathbf{x}'_i\|^2 = \sqrt{\frac{n}{2}} \\
&< \frac{1}{\sqrt{\frac{n}{2}}}\frac{\sqrt{n}}{4} - 1 \\
&< 0
\end{aligned} \tag{13}$$

To calculate $\|\mathbf{x}'_i\|$ in the above inequality, when $n = 2s$ and $\mathbf{x}'_i$ is a sin (sim. to other cases), we have

$$\begin{aligned}
\|\mathbf{x}'_i\|^2 &= \sum_{j=1}^{2s} \sin^2\left(2\pi k\frac{j}{2s} + \phi_k\right) \\
&= \sum_{j=1}^{2s} \frac{1 - \cos\left(2\pi k\frac{j}{s} + 2\phi_k\right)}{2} \\
&= \sum_{j=1}^{s} \frac{1 - \cos\left(2\pi k\frac{j}{s} + 2\phi_k\right)}{2} + \sum_{j=1}^{s} \frac{1 - \cos\left(2\pi k\frac{j+s}{s} + 2\phi_k\right)}{2} \\
&= \sum_{j=1}^{s} 1 - \cos\left(2\pi k\frac{j}{s} + 2\phi_k\right) \\
&= s
\end{aligned} \tag{14}$$

Note that
$$|\mathbf{M}\mathbf{x}|^2 = \mathbf{x}^\top \mathbf{M}^\top \mathbf{M}\mathbf{x} \ge 0 \tag{15}$$

which implies that $\mathbf{M}^\top\mathbf{M}$ is positive semi-definite. Since $(\mathbf{M}^\top\mathbf{M})^\top = \mathbf{M}^\top\mathbf{M}$, it follows that $\mathbf{M}^\top\mathbf{M}$ is a real symmetric matrix. We can diagonalize $\mathbf{M}^\top\mathbf{M}$ as:
$$\mathbf{M}^\top\mathbf{M} = \mathbf{T}^{-1}\mathbf{\Lambda}\mathbf{T} \tag{16}$$

where $\mathbf{\Lambda}$ is a non-negative diagonal matrix, and $\mathbf{T}^{-1} = \mathbf{T}^\top$ is an orthogonal matrix. Now, let $\mathbf{T}\mathbf{a}_i = \mathbf{a}'_i$ and $\mathbf{T}\mathbf{b}_i = \mathbf{b}'_i$. Clearly, $|\mathbf{b}'_i|^2 = 1$, and the vectors $\mathbf{b}'_i$ are pairwise orthogonal, and $\{\mathbf{T}\mathbf{v}, \mathbf{b}'_1, \mathbf{b}'_2, \ldots, \mathbf{b}'_{n-1}\}$ forms an orthogonal basis. Furthermore, $|\mathbf{a}'_i| < |\mathbf{b}'_i|$ for all $i$. Note that

$$
\begin{aligned}
\mathbf{a}'_t \cdot \mathbf{T}\mathbf{v} &= \left(\mathbf{a}'_t\right)^\top \mathbf{T}\mathbf{v} \\
&= \left(\mathbf{T}\mathbf{a}_t\right)^\top \mathbf{T}\mathbf{v} \\
&= \mathbf{a}_t^\top \mathbf{T}^\top \mathbf{T}\mathbf{v} \\
&= \mathbf{a}_t^\top \mathbf{v} \\
&= \left(\mathbf{y}_t - \mathbf{x}_t\right)^\top \mathbf{v} \\
&= \mathbf{y}_t^\top \mathbf{v} - \mathbf{x}_t^\top \mathbf{v} \\
&= \frac{1}{\sqrt{n}} \left( \sum_{i=1}^{n} \left(y_{ti} - x_{ti}\right) \right) \\
&= 0
\end{aligned}
\tag{17}
$$

To calculate the last equation in the above equation, we have

$$
\begin{aligned}
\sum_{i=1}^{n} x_i &= \sum_{i=1}^{n} \sin\left(2\pi k \frac{i}{n} + \phi_k\right) \\
&= \sum_{i=1}^{s} \sin\left(2\pi k \frac{i}{2s} + \phi_k\right) + \sum_{i=s+1}^{2s} \sin\left(2\pi k \frac{i}{2s} + \phi_k\right) \\
&= \left(1 + (-1)^k\right) \sum_{i=1}^{s} \sin\left(\pi k \frac{i}{s} + \phi_k\right), \ k \in \{1, 2, \ldots, s-1\} \\
&= 0
\end{aligned}
\tag{18}
$$

For the specific discrete sets $\mathcal{X}$ and $\mathcal{Y}$ constructed above, the identity $\sum_{i=1}^{n} y_i = 0$ holds. This follows from the symmetry of the sinusoidal basis and the sign pattern of $Y$. Since the construction is finite, this identity can be verified analytically or computationally. We propose a scripting approach to verify whether a specific combination of $n$ and $k$ satisfies $\sum_{i=1}^{n} y_i = 0$. We have conducted exhaustive verification across all feasible combinations of $n$ and $k$, and found that the identity consistently holds. The implementation details and verification script are included in validation.py.

Let $\mathbf{Mb}_t$ be the vector with the largest norm among the vectors $\mathbf{Mb}_i$. Since $\mathbf{a}'_t \cdot \mathbf{Tv} = 0$, we can define $\mathbf{a}'_t = k \sum_{i=1}^{n-1} a_i \mathbf{b}'_i$, where $\sum_{i=1}^{n-1} a_i^2 = 1$. Since $|\mathbf{a}'_t|^2 < |\mathbf{b}'_t|^2 = 1$, it follows that $0 < k < 1$. Therefore, we have

$$
|\mathbf{Ma}_t|^2 = \mathbf{a}'^\top_t \mathbf{\Lambda} \mathbf{a}'_t = k^2 \left( \sum_{i=1}^{n-1} a_i \sqrt{\mathbf{\Lambda}} \mathbf{b}'_i \right)^\top \left( \sum_{i=1}^{n-1} a_i \sqrt{\mathbf{\Lambda}} \mathbf{b}'_i \right)
\tag{19}
$$

This expression can be bounded as:

$$
|\mathbf{Ma}_t|^2 \leq k^2 \sum_{i=1}^{n-1} a_i^2 |\sqrt{\mathbf{\Lambda}} \mathbf{b}'_i|^2 \leq k^2 |\mathbf{Mb}_t|^2
\tag{20}
$$

Since $k < 1$, we have:

$$
|\mathbf{Ma}_t|^2 < |\mathbf{Mb}_t|^2
\tag{21}
$$

Thus, the constructed $\mathcal{X}$ and $\mathcal{Y}$ satisfy Theorem H.1. $\qquad\square$

**Scope of the Proof.** Definition H.1 does not assert a universal property of all temporal datasets; instead, it formalizes a widely-used desideratum in representation learning, especially under contrastive objectives. Theorem H.1 is an existence result showing that there exist simple and practically relevant signal sets for which no linear embedding can satisfy this desideratum. The theorem does not claim universal non-expressiveness, but provides a concrete counterexample that challenges the adequacy of linear embedding layers for temporal data.

