# OpenReview forum: "Why Attention Fails: The Degeneration of Transformers into MLPs in Time Series Forecasting"
_ICLR.cc/2026/Conference — Submitted to ICLR 2026_

### Official Review · Reviewer_QRm4 · 2025-10-24

**Soundness:** 1
**Presentation:** 2
**Contribution:** 1
**Rating:** 2
**Confidence:** 4

**Summary:**

The paper investigates why time-series Transformers (TSTs) often underperform simple baselines in forecasting. The main claim is that, in many widely-used TST variants (e.g., PatchTST, iTransformer), attention contributes little to accuracy and the backbone effectively degenerates into an MLP/FFN.

**Strengths:**

1. A simple diagnostic offers a unified way to understand attention’s functional role in TSTs.
2. The toy state-machine dataset provides interpretable ground truth about inter-patch dependencies

**Weaknesses:**

1. Replacement/perturbation experiments show robustness to disabling attention, but they do not establish why attention underperforms relative to FFN, nor isolate which design choices (tokenization stride/length, residual pathways, normalization, optimizer/sharpness) make attention redundant. Without causal attribution, the paper does not show understanding beyond “attention might not matter”.
2. Disabling attention changes model capacity and computation. It remains unclear whether comparable performance arises because attention is redundant or because remaining layers dominate the capacity. When attention is removed or replaced, the model’s parameter count, compute, and capacity all change. The paper does not control for these factors, so it is impossible to attribute performance parity to attention’s irrelevance rather than architectural imbalance.
3. The “linear embedding hypothesis” is plausible but lacks formal backing. The paper makes a strong conceptual claim without proof or assumptions.
4. Present results on shared-axis figures, making small changes look negligible. Have you tried confidence intervals or statistical significance tests, so “no difference” may be an artifact of noise?
4. The title and narrative (“Why Attention Fails”) imply a universal conclusion, but the study covers only a narrow class of models (mostly TST-style architectures with linear embeddings). It omits settings where attention could matter (e.g., irregular sampling, exogenous covariates, long-horizon dependencies). The authors do not demonstrate any counterexamples or boundary conditions where the attention module could help.
5. The toy dataset is interesting but disconnected from the real-world findings. It visualizes “attention not focusing correctly,” but the authors did not bridge the toy results to why real models fail.
6. The writing is uneven and sometimes confusing, with grammatical errors, and vague reasoning (“we think linear embedding is bad,” “attention does not help much”).

**Questions:**

1. Are the capacity-matched MLP/TSMixer baselines trained under identical compute and regularization?
2. Does introducing a nonlinear embedding layer (e.g., small CNN or random Fourier features) restore the usefulness of attention?
3. Are there datasets or regimes (e.g., long-context forecasting, irregular sampling, exogenous signals) where attention demonstrably helps?
4. Would you consider using per-dataset metric deltas or significance tests to confirm that “no difference” results are statistically meaningful?

---

> ### Author Response · Authors · 2025-11-21
> **Point-by-Point Rebuttal to Reviewer Critiques**
>
> **Q1 Without causal attribution, the paper does not show understanding beyond “attention might not matter”.**
>
> We have updated the PDF. In Appendix G, we provide our mathematical proof showing that a linear layer cannot provide a good latent representation. In Appendix H, we provide an Oracle Embedding approach, which confirms that a good latent representation allows the attention mechanism to function properly.
>
> Combining this with the large number of experiments throughout the paper, the failure of representation learning is the cause of the degeneration of attention. This goes beyond most other papers that only explore the idea that “attention might not matter.”
>
> Therefore, this paper is novel in that it bypasses the traditional research paradigm of time-series Transformers and proposes a new direction focusing on representation learning.
>
> ---
>
> **Q2 The study covers only a narrow class of models.**
>
> In Tables 8–23, we present results for Transformer models focusing on different optimization directions, including patch-wise, channel-wise, TSFMs, new optimizers, and so on.
>
> You may suggest other representative open-source models from domains you consider important (e.g., irregular sampling, exogenous covariates, long-horizon dependencies). We will run additional experiments on them and return the results to you as quickly as possible. We have previously conducted informal experiments on multiple different classes of Transformers, and the results were consistent with those reported in this paper. We expect that the models you recommend will very likely also exhibit degeneration.
>
> ---
>
> **Q3 The authors do not demonstrate any counterexamples or boundary conditions where the attention module could help.**
>
> We have updated the PDF. In Appendix H, we provide an Oracle Embedding approach, which confirms that a good latent representation allows the attention mechanism to function properly.
>
> ---
>
> **Q4 The authors did not bridge the toy results to why real models fail.**
>
> In Appendix H, the toy dataset results clearly show that the failure of representation learning is the cause of the degeneration of attention. In Section 5 and Appendix A, we explicitly discuss why real models fail: the characteristics of time-series data make representation learning difficult, which in turn causes the degeneration of the attention mechanism.
>
> ---
>
> **Q5 Does introducing a nonlinear embedding layer (e.g., small CNN or random Fourier features) restore the usefulness of attention?**
>
> According to Table 4, existing mainstream implementations of embedding layers cannot restore the usefulness of attention.
>
> ---
>
> **Q6 Would you consider using per-dataset metric deltas or significance tests to confirm that “no difference” results are statistically meaningful? Present results on shared-axis figures, making small changes look negligible.**
>
> In Tables 8–19, we provide extensive experiments across different datasets, different models, and different metrics, including statistical indicators. All results are presented as numbers rather than figures, thus avoiding the issue that “no difference may be an artifact of noise.”
>
> ---
>
> **Q7 Are the capacity-matched MLP/TSMixer baselines trained under identical compute and regularization?**
>
> Yes.
>
> ---
>
> **Q8 Disabling attention changes model capacity and computation.**
>
> In lines 197–205, we discuss the potential effects of changes in model capacity and computation. For the experiments in Section 3.3, we adjust the hidden dimensions corresponding to different patch lengths in the code to keep the number of parameters and computation as consistent as possible; details can be found in the anonymous repository we have provided. In fact, because most experiments only lead to a decrease in computation, and performance remains stable after the decrease, these changes in capacity and computation do not undermine the conclusions but instead further strengthen them.
>
> ---
>
> **Q9 The “linear embedding hypothesis” is plausible but lacks formal backing.**
>
> We have updated the PDF. In Appendix G, we now provide our proof. Please refer to that section.

---

> > ### Comment · Reviewer_QRm4 · 2025-11-22
> >
> > The counterexample is still not rigorously established because:
> >
> > 1. The vectors in $\mathcal{X}$ are not orthogonal in general.
> >
> > The paper says "it is easy to observe that the vectors in $\mathcal{X}$ are mutually orthogonal," but this is not trivial and not correct for arbitrary per-vector phases. The missing piece is a clear constraint on the phase parameters $\phi_k$ and $\psi_k$. One possible condition would be to assume that $\phi_k - \psi_k$ is a multiple of $\pi$ for every $k$. Without such constraints, the orthogonality claim does not hold.
> >
> > 2. Another condition on the phases is needed to make $\mathcal{X}$ symmetric.
> >
> > If I understand correctly, the goal is to define $a_i$ as the cross-class difference and $b_i$ as the within-class difference. For this to be valid, the set $\mathcal{X}$ needs to be closed under taking negatives; otherwise, $-x_i$ is not in the set, and the interpretation of $b_i = x_i - (-x_i)$ as a within-class distance breaks. Right now, the paper only lists the cos/sin forms once for each frequency, with no explicit assumptions on phases, so the symmetry is not guaranteed.
> >
> > 3. For the claim that the $y_i$'s sum to zero, the current assumptions are not enough.
> >
> > So far, nothing in the paper guarantees that the sign patterns have zero sum. Orthogonality of the $x_i$'s does not imply that $\sum y_i = 0$. If the basis were a very specific and carefully aligned trigonometric basis, the identity might hold, but with the general phase freedom allowed in the definitions, it does not follow. The additional issue here is that, even though the vector space is finite-dimensional, the set of admissible vectors is infinite because the phases $\phi_k$ and $\psi_k$ vary continuously. It is simply impossible to test all possible phase pairs, so computational checking cannot replace a proof.
> >
> > ***
> >
> > On the main claim of the paper:
> >
> > My understanding is that the results mainly show that attention is redundant, or at least not needed, in many time-series forecasting settings, and that FFNs dominate. But this does not convincingly show that attention "failed," nor that intrinsic representation learning failed. To me, this looks more like task-level redundancy, not degeneration of attention. The title made me expect something broader: For example, evidence that attention cannot operate even in situations where it should help. But that kind of failure is never demonstrated. Instead, the results are consistent with what we would expect: most time-series datasets are highly redundant and low-rank, and in such cases attention naturally becomes unnecessary. In my view, this is not a negative result; it actually suggests an opportunity to design architectures specifically for time-series, instead of directly reusing NLP/ViT designs.

---

> ### Author Response · Authors · 2025-11-23
> **Point-by-Point Rebuttal to Reviewer Critiques**
>
> **Q10 The paper does not convincingly show that attention "failed," nor that intrinsic representation learning failed.**
>
> Thank you for the question. In our responses to other reviewers, we have provided the results in Appendix H. The results clearly show that if proper representation learning is available, attention can function effectively, which strongly supports our argument.
>
> ---
>
> **Q11 Attention naturally becomes unnecessary, which suggests an opportunity to design architectures specifically for time-series, instead of directly reusing NLP/ViT designs.**
>
> Your point is very insightful. In fact, your perspective is not in conflict with our conclusion. The findings of this paper are intended to guide new architecture designs toward placing greater emphasis on representation learning for time series. As discussed in Appendix A, due to differences in data characteristics, the difficulty of representation learning in time-series tasks is not the same as in NLP/CV, and therefore we need designs different from NLP/ViT. This is fully aligned with your expectation.
>
> ---
>
> **Q12 The vectors in $_\mathcal{X}$ are not orthogonal in general.**
>
> Thank you for your patience and careful examination. We have corrected this in Equation (9).
>
> ---
>
> **Q13 Another condition on the phases is needed to make $_\mathcal{X}$ symmetric.**
>
> Thank you for the question. We have clarified this in lines 1455–1456.
>
> ---
>
> **Q14 For the claim that the $y_i$'s sum to zero, the current assumptions are not enough.**
>
> Thank you for your patience and careful examination. We corrected the phase setting in Q10. Moreover, Theorem G.1 only needs to find a case where Definition G.1 is not satisfied. For the claim that the $y_i$'s sum to zero, computational checking does not need to cover all possible scenarios. Nonetheless, we still covered a large number of parameter combinations in our script, and the results matched our expectations.

---

### Official Review · Reviewer_6ebT · 2025-10-27

**Soundness:** 2
**Presentation:** 3
**Contribution:** 2
**Rating:** 4
**Confidence:** 3

**Summary:**

The paper argues that Transformer architectures often degenerate into simple MLP when applied to time series forecasting. The attention mechanism fails to function as intended, rendering the model effectively equivalent to an MLP.

**Strengths:**

* Clear, Counter-intuitive Finding: It identifies and systematically demonstrates a surprising phenomenon: Transformers effectively function as MLPs in time series forecasting, making a strong challenge to the field.

* Interpretable Analysis: The design of a custom toy dataset allows for a clear inspection of attention failure, moving beyond just performance metrics on complex, black-box real-world data.

* Broad Validation: The findings are shown to hold across multiple modern Transformer architectures (PatchTST, iTransformer) and even large time series foundation models, proving the issue is widespread.

* Actionable Insight: It moves beyond just identifying a problem to pinpointing a root cause: the failure of representation learning and the inadequacy of simple linear embeddings, giving a clear direction for future work.

**Weaknesses:**

1. "Degeneration" is Loosely Defined: The term "degeneration into an MLP" is powerful but imprecise. A true MLP on a flattened sequence would still have weights connecting all input points. In their experiments (like Zero attention), the model still processes tokens independently through the FFN and only mixes them at the final linear layer. This is a specific type of architecture, not a standard MLP. The claim is more accurately that the model becomes token-wise independent until the very end, which is a crucial distinction.

2. The Primacy of the Embedding Layer is an Assertion, Not a Proven Cause: The paper identifies linear embedding as the root cause but provides only correlative, not causative, evidence. They show that linear embedding is ineffective, but their other experiments (Table 4) show that other embedding methods (Conv, MLP) also fail to prevent attention degeneration. This strongly suggests the problem is deeper and more fundamental than just the choice of a linear layer. The core issue might lie in the nature of time series data itself or the overall training objective, which the paper only briefly touches on in the appendix.

3. Insufficient Distinction Between "Not Necessary" and "Not Working": The experiments brilliantly show that attention is not necessary for good performance on these tasks. However, this is not the same as proving it is "not working." It's possible that in the standard model, the attention mechanism is learning meaningful patterns, but the FFN is learning a redundant, more efficient pathway. Ablating attention forces the model to rely solely on the FFN, which, it turns out, is sufficient. This is a subtle but important difference in interpretation.

4. Lack of Analysis on Learned Representations: The paper claims the latent space is poorly structured but provides limited direct analysis of the learned token embeddings. A more convincing case could be made by using probes or similarity measures to show that the representations for tokens that should be related (e.g., similar events in the time series) are not closer in the latent space than unrelated tokens.

5. The section labeled "Theoretical Analysis" (Section 5.1) is more of a conceptual discussion or a position statement than a rigorous mathematical analysis. It lacks the formal definitions, theorems, proofs, and mathematical modeling that would qualify as a theoretical foundation.

**Questions:**

Please see Weaknesses.

---

> ### Author Response · Authors · 2025-11-21
> **Point-by-Point Rebuttal to Reviewer Critiques**
>
> **Q1 “Degeneration” is Loosely Defined.**
>
> “Degeneration of the attention mechanism” means that the attention mechanism does not contribute to the model’s performance, causing the model before the final flatten head to become a stack of residual MLPs.
>
> ---
>
> **Q2 The Primacy of the Embedding Layer is an Assertion, Not a Proven Cause. The paper provides only correlative, not causative, evidence.**
>
> We have updated the PDF. In Appendix H, we provide an Oracle Embedding approach, which confirms that a good latent representation allows the attention mechanism to function properly. This demonstrates the necessity of representation learning and validates the primacy of the embedding layer.
>
> ---
>
> **Q3 “Theoretical Analysis” (Section 5.1) is more of a conceptual discussion or a position statement than a rigorous mathematical analysis. & Lack of Analysis on Learned Representations.**
>
> We have updated the PDF. Appendix G now includes our mathematical proofs. Please refer to that section.
>
> ---
>
> **Q5 Insufficient Distinction Between “Not Necessary” and “Not Working.”**
>
> For this paper, there is no need to distinguish between “not necessary” and “not working.” Regardless of whether the attention mechanism is “not necessary” or “not working,” both imply that the attention mechanism must provide a meaningful performance improvement.
>
> If the attention mechanism does not deliver a substantial and commensurate improvement despite introducing a large number of parameters and FLOPs, then we consider it to be failing. If an MLP alone can achieve the same effect, then we do not need the attention mechanism, because attention introduces quadratic complexity. Therefore, even if the attention mechanism is learning meaningful patterns, that is still insufficient—it needs to perform better.

---

### Official Review · Reviewer_crtu · 2025-10-30

**Soundness:** 3
**Presentation:** 4
**Contribution:** 3
**Rating:** 6
**Confidence:** 4

**Summary:**

This paper presents a crucial and timely diagnostic study investigating why Transformer-based models (TSTs) consistently fail to outperform simple linear models in Time-Series Forecasting (TSF), a phenomenon widely observed since the DLinear paper.

The authors' core contribution is the compelling demonstration that TSTs often degenerate into simple MLPs. Through a meticulous series of experiments, they show that the self-attention mechanism becomes functionally inert. The paper traces this failure back to a single root cause: the ineffectiveness of the standard linear embedding layer. This layer, adopted from NLP/CV, fails to project time-series data into a meaningful latent space, leaving the attention mechanism with no structured information to operate on.

**Strengths:**

1. This is an exceptionally strong diagnostic paper that provides immense value to the TSF community, even without proposing a new SOTA model.

- Meticulous and Rigorous Experimental Design: The paper's primary strength is its evidence. The authors don't just ablate attention; they show its progressive degeneration through multiple, clever experiments:
- Attention Replacement (Sec 3.1): Replacing attention with Zero, Identity, and Mean matrices proves the model's performance is not derived from context-aware attention.
- Module Perturbation (Sec 3.2): Demonstrates that perturbing the FFN module drastically harms performance, while perturbing the attention module does almost nothing.

Patch Length Variation (Sec 3.3): Shows that performance is stable even when the model is degenerated to a single-token MLP by increasing patch length.

2. The comparison with ViT (Sec 3.4) is perhaps the most powerful argument. By showing that ViT breaks when positional encoding is removed while PatchTST does not, the authors masterfully isolate the problem. It's not the Transformer architecture that's flawed, but its application to poorly represented time-series data.

3. This paper provides a clear, evidence-based answer to a puzzle that has occupied the TSF community. It will save many researchers from the flawed path of "building a better attention mechanism" and correctly refocuses the field on the true bottleneck: representation learning for time-series data.

**Weaknesses:**

1. Contribution is Diagnostic, Not Prescriptive: The most significant weakness is that the paper provides an excellent diagnosis but no validated cure. It "closes the loop" on the problem but not on the solution. While Appendix D speculates on non-linear encoders (like VQ-VAE), the paper would have been far more complete if it had demonstrated a fix.

2. Missed Opportunity on the Toy Dataset: The authors created the perfect testbed (the toy dataset) to validate a solution. A key missed opportunity was not applying a simple non-linear encoder (e.g., an MLP-based embedder) to this dataset. Showing that a better embedding re-activates attention (i.e., makes it focus on the "event" patch in Fig 7) would have transformed this from a great diagnostic paper to a groundbreaking complete paper.

3. Novelty of the Problem is Low: As the authors acknowledge, the problem that TSTs underperform linear models is well-known. The novelty lies entirely in the depth and clarity of the analysis, not in the discovery of the problem itself.

**Questions:**

This is an insightful paper, and I have a few questions to better understand the boundaries of its conclusions.

1. Closing the Loop on the Toy Dataset: Your toy dataset was the most compelling part of the analysis. Have you attempted to replace the linear embedding with a simple non-linear encoder (e.g., a small MLP or 1D-CNN) on this specific dataset? If so, were you able to observe the attention mechanism becoming "re-activated" and focusing on the critical "event" patches as one would expect? This seems like a crucial missing experiment to fully validate your hypothesis.

2. ViT's Success: Data Density vs. Architecture? You argue that ViT succeeds because its early FFNs perform representation learning (Sec 5.3), which TSTs fail to do even with more blocks (Fig 19, 20). Do you attribute this fundamental difference to (a) the nature of the data (i.e., image patches have a much higher "information density" than time-series patches) or (b) a subtle architectural difference in how TSTs are structured (e.g., the final linear head)?

3. Implications for Large Foundation Models: Your analysis is based on training models on individual datasets. However, large models like ViT learned rich representations because they were trained on massive, diverse datasets (like ImageNet). Do you believe your findings still apply to large-scale Time-Series Foundation Models (TSFMs) like Moirai or TimesFM? Or is it possible that "scale" (i.e., pre-training on millions of time series) can overcome the poor linear embedding and teach the model a meaningful latent space, thereby re-activating the attention mechanism?

---

> ### Author Response · Authors · 2025-11-21
> **Point-by-Point Rebuttal to Reviewer Critiques**
>
> **Q1 The paper does not close the loop on the toy dataset.**
>
> We have updated the PDF. In Appendix H, we provide an Oracle Embedding approach, which confirms that a good latent representation allows the attention mechanism to function properly. In the experiment, as you mentioned, the attention mechanism becomes “re-activated” and focuses on the critical “event” patches as one would expect.
>
> ---
>
> **Q2 Novelty of the Problem is Low: the problem that TSTs underperform linear models is well-known.**
>
> Our work aims to propose a new research direction, revealing that we should shift our focus to representation learning rather than optimizing the attention mechanism itself.
>
> Previous works have also studied the ineffectiveness of attention, but these works did not provide fundamental guidance for subsequent research. A considerable portion of new works still focus on optimizing model architectures or scaling models and datasets. However, the attention mechanisms in these newer works remain degenerated. We hope this paper can draw more attention to representation learning.
>
> Therefore, this paper is novel because it bypasses the traditional research paradigm of time-series Transformer studies and proposes a new direction centered on representation learning.
>
> ---
>
> **Q3 ViT’s Success: Data Density vs. Architecture?**
>
> This fundamental difference attributes to the nature of the data. As we point out in Appendix A, CV and TS data differ significantly in many aspects, such as the input data size corresponding to a single token, the scaling ratio from input data to hidden dimensions, the generalizability across datasets, and the degree of distributional variation within a single dataset. In addition, data augmentation in CV differs from TS—for example, a mirrored image of a dog is still a dog, whereas time-series data cannot be flipped along the temporal axis.
>
> We have experimented with different variants of PatchTST, such as modifying the final linear head (as you mentioned), and the results are consistent with the other experiments in the paper. Therefore, we do not believe the cause is architectural.
>
> ---
>
> **Q4 Do the findings still apply to large-scale Time-Series Foundation Models (TSFMs) like Moirai or TimesFM?**
>
> For TSFMs, the degeneration of attention still occurs. We tested Lag-Llama, Timer, and Moirai, and the results in Tables 20, 22, and 23 are consistent with the other experiments in the main text.
>
> As stated in lines 666–678, the distributional differences between different time-series datasets are enormous. For example, it is difficult to determine whether experiences from an electricity dataset with strong periodicity can transfer to a high-noise, non-periodic financial dataset. It is also difficult to determine whether the patterns across different datasets are non-contradictory. In fact, for the toy dataset, it is easy to design mutually contradictory state machines, and such contradictory state machines cannot be trained within the same model.
>
> In contrast, in the CV domain, the world knowledge contained in different datasets is largely consistent. Lighting, texture, refraction, reflection, and other physical properties are universal and consistent, making it possible to train a single model on very large datasets or across different domains.

---

> > ### Comment · Reviewer_crtu · 2025-11-24
> >
> > While my initial positive rating was primarily motivated by the paper's detailed empirical analysis, the review process and the subsequent addition of Appendix G have led me to reassess the requirements for this work. I have come to the conclusion that for the paper's core argument regarding the fundamental failure of linear embeddings to hold, a rigorous mathematical justification is indispensable, not just supplementary.
> >
> > Unfortunately, the attempt to provide this theoretical grounding in Appendix G falls short. I agree with Reviewer QRm4 that the provided "Mathematical Proof" lacks the necessary rigor to support the claims.
> > Specifically:
> > 1. Unguaranteed Orthogonality: The assumption that vectors in $\mathcal{X}$ are mutually orthogonal is invalid without strict constraints on the phase parameters ($\varphi_k, \psi_k$)
> >
> > 2. Unjustified Symmetry: The proof assumes $\mathcal{X}$ is closed under negation ($-x_i \in \mathcal{X}$) to define the intra-class distance $b_i$3, which is not mathematically proven for general cases.
> >
> > 3. Reliance on Scripting: Crucially, the proof substitutes analytical derivation with a "scripting approach" and code execution (validation.py) to verify $\sum y_i = 0$. Empirical verification is insufficient for a formal mathematical theorem.
> >
> > Since the foundational inequalities depend on these unproven assumptions, the theorem is mathematically unsound.
> > A flawed proof undermines the reliability of the paper, necessitating a score adjustment.

---

> > > ### Author Response · Authors · 2025-11-27
> > > **Point-by-Point Rebuttal to Reviewer Critiques**
> > >
> > > Thank you for your careful and thorough review. Your rigorous attitude toward the proofs has helped us improve the rigor of our paper. We have corrected the detailed issues in the proofs, and since the proofs are not the core content, we hope to have the opportunity to discuss the core experimental results with you.
> > >
> > > Our central claim is that **mainstream embedding** methods lead to a failure of representation learning, which in turn causes the degeneration of the attention mechanism — not the claim of “the fundamental failure of **linear embeddings**.” Therefore, our most important experiments are those on the toy dataset, in particular the experiments reported in Appendix H that were added in response to your Q1. Please review them in detail.
> > >
> > > Regarding the concerns raised by Reviewer QRm4, we have addressed them; please read the revised paper rather than repeat QRm4’s points. Specifically, we have updated the phase settings to ensure orthogonality; $\chi$ is used to represent an orthogonal basis later, so we did not include $-\mathbf{x}$ (this is not an issue of phase settings); when computing $b_i$, it is sufficient to use samples from the same class; and concerning the reliance on scripting, Theorem G.1 only requires finding a counterexample to Definition G.1 and does not require exhaustively enumerating all scenarios.
> > >
> > > Thank you again for your review.

---

### Official Review · Reviewer_urWk · 2025-10-31

**Soundness:** 3
**Presentation:** 3
**Contribution:** 3
**Rating:** 6
**Confidence:** 4

**Summary:**

The paper titled “Why Attention Fails: The Degeneration of Transformers into MLPs in Time Series Forecasting ”  investigates why Transformer architectures do  not have a clear advantage in time series forecasting and often shown to underperform simpler baselines. The main result (found empreically) is that in time-series Transformers (TSTs), the attention blocks often degenerate into MLPs, as the attention mechanism fails to function as intended. Extensive experiments, including attention replacement and perturbation, validate that the Feed-Forward Network (FFN) is the dominant player in performance, while the contribution of the attention module is minimal. The authors empirically argue that the problem stems from ineffective linear embedding methods, that hinder the attention mechanism from capturing crucial inter-token dependencies in time series data.

**Strengths:**

The paper observes an interesting phenomenon in that it identifies and substantiates the critical finding that Transformer blocks in Time Series Transformers (TSTs) frequently degenerate into simple Multi-Layer Perceptrons (MLPs).

Experimental validation is multifaceted and extensive and has managed to cover many aspects such as attention replacement, perturbation experiments, varying patch length among others

The findings are validated across a variety of architectures, including recent patch-wise and channel-wise Transformer models as well.

**Weaknesses:**

The theoretical analysis regarding linear embedding is largely an assertion or intuition that the isomorphic linear transformation of a full-rank linear layer cannot map a complex time series manifold into a "well-structured latent space". This is presented without providing any formal mathematical proofs, theorems, or guarantees (e.g., lower bounds, expressivity analysis) that rigorously demonstrate why this structural mapping is fundamentally impossible or sub-optimal in a general sense.

The primary conclusion is that the issue lies with poor representation learning due to ineffective linear embeddings.  But it begs the question, what would be a good embedding then? On this front paper explicitly does not include a thorough discussion or evaluation of alternative embedding strategies, except in a discussion section

The key experiments focus o attention replacements (zero, identity, mean) or simple perturbations (noise/attenuation). While effective at showing the lack of necessity of attention, this approach does not entirely rule out the possibility that a different, better-designed attention mechanism could be learned.

**Questions:**

na

---

> ### Author Response · Authors · 2025-11-21
> **Point-by-Point Rebuttal to Reviewer Critiques**
>
> **Q1 The theoretical analysis regarding linear embedding is largely an assertion or intuition without providing any formal mathematical proofs.**
>
> We have updated the PDF and provided our proof in Appendix G of the paper. Please kindly refer to it.
>
> ---
>
> **Q2 What would be a good embedding?**
>
> A good embedding should be able to capture the semantic characteristics of the input data, allowing the dot-product between (q) and (k) in the attention mechanism to reflect similarities between different semantic directions in a high-dimensional semantic space. We have updated the PDF and added an oracle embedding method in Appendix H, which demonstrates that a good latent representation indeed enables the attention mechanism to function properly.
>
> According to the results in Table 4, current mainstream embedding methods are ineffective. We have not yet identified any new embedding strategies.
>
> ---
>
> **Q3 The key experiments do not entirely rule out the possibility that a different, better-designed attention mechanism could be learned.**
>
> We evaluate whether the attention mechanism is effective based on the model’s performance. Even if a different, better-designed attention mechanism is learned, it would still be meaningless if it does not yield a positive performance gain.

---

### Meta-Review · Area_Chair_9wcW · 2026-01-07

**Summary:**

The paper investigates why time-series Transformers (TSTs) often degenerate into simple MLPs when applied to time series forecasting. The main result (found empirically) is that in time-series Transformers (TSTs), the attention blocks often degenerate into MLPs, as the attention mechanism fails to function as intended. Extensive experiments, including attention replacement and perturbation, validate that the FFN is the dominant player in performance, while the contribution of the attention module is minimal. This paper empirically traces this failure back to a single root cause, i.e., the problem stems from the ineffectiveness of the standard linear embedding layer, which hinders the attention mechanism from capturing crucial inter-token dependencies in time series data. Note that this layer, adopted from NLP/CV, fails to project time-series data into a meaningful latent space, leaving the attention mechanism with no structured information to operate on. Extensive and comprehensive evaluations across a wide range of architectures, datasets, and ablation settings demonstrate that the self-attention module frequently contributes little to performance, with the FFN dominating model behavior. Besides, the toy dataset and the attention replacement/perturbation studies are particularly insightful and provide strong interpretability and community value.

Reviewers acknowledged that i) this insightful paper observes an interesting phenomenon and ii) strong empirical evaluations with interpretable analysis. During the rebuttal, to address several reviewers’ concerns regarding causality, the authors have provided new experiments (Appendix H) to demonstrate that when a suitable latent representation is provided, the attention mechanism can indeed be reactivated. This result clarifies that the observed degeneration is closely tied to representation learning rather than an inherent flaw of attention itself.

However, significant concerns remain regarding the rigor of the theoretical analysis. The mathematical arguments in Appendix G are central to the paper’s strongest claims about the limitations of linear embeddings. However, the theoretical conclusions appear overstated relative to the evidence. As pointed out by Reviewer QRm4, the title and narrative (“Why Attention Fails”) of this paper imply a universal conclusion, but the study covers only a narrow class of models. The empirical findings are robust and valuable, but the narrative would benefit from a more cautious framing that distinguishes between attention being redundant in common TSF settings and attention fundamentally “failing”.

In summary, this paper is somewhat weakened by over-ambitious theoretical claims. I lean towards rejection in its current form and encourage the authors to substantially revise the paper.

**Reviewer Concerns:**

The rebuttal successfully addressed several key concerns raised by the reviewers. In particular, it responded well to questions from Reviewers urWk, crtu (initial review), and 6ebT regarding whether attention can function at all, by adding the Oracle Embedding experiments (Appendix H), which provide convincing evidence that a suitable latent representation can re-activate attention and clarify the causal role of representation learning. The rebuttal also adequately addressed concerns about experimental breadth, capacity mismatch, and the connection between toy and real-world datasets, as raised by Reviewers crtu and QRm4, by adding results on additional architectures and large-scale TSFMs and by clarifying experimental controls.

However, significant concerns remain outstanding regarding the rigor of the theoretical analysis in Appendix G. The objections raised by Reviewers QRm4 and crtu in the later stages, i.e., specifically about unproven assumptions, reliance on special constructions, and the use of empirical scripting in place of formal proofs, were not fully resolved and continue to undermine the strongest theoretical claims. Additionally, concerns about over-strong framing (e.g., “attention fails” versus “attention is often redundant”) remain only partially addressed.

**Reviewer Scores:**

Reviewer urWk would likely have maintained or slightly increased their score. Their main concerns were about the lack of theoretical justification and the absence of discussion of better embeddings. Both were at least partially addressed by the added Appendix G and, more convincingly, Appendix H, which aligns well with their emphasis on empirical validation and broad analysis.

Reviewer crtu would likely have lowered their score slightly after fully participating in the discussion. While they initially viewed the paper very positively, their later comments show growing dissatisfaction with the rigor of the theoretical proof and discomfort with the paper’s strongest claims.

Reviewer 6ebT would probably have remained near their original score. The rebuttal clarified the definition of “degeneration” and strengthened the causal argument via Oracle Embeddings, but their deeper concern, i.e., the embedding layer is asserted as the root cause without definitive proof, was only partially addressed.

Reviewer QRm4 would almost certainly have kept their score at the rejection level. Their objections are fundamentally about theoretical rigor, causal attribution, and over-claiming in the paper’s framing. He/She explicitly stated that the counterexamples and proofs remain unsound and that the results are better interpreted as attention being redundant rather than failing.

---

### Decision · Program_Chairs · 2026-01-26

Reject